# Upweighting Easy Samples in Fine-Tuning Mitigates Forgetting

**Sunny Sanyal** [* 1]   **Hayden Prairie** [* 1]   **Rudrajit Das** [* 2]   **Ali Kavis** [* 1]   **Sujay Sanghavi** [1]

## Abstract

Fine-tuning a pre-trained model on a downstream task often degrades its original capabilities, a phenomenon known as "catastrophic forgetting". This is especially an issue when one does not have access to the data and recipe used to develop the pre-trained model. Under this constraint, most existing methods for mitigating forgetting are inapplicable. To address this challenge, we propose a *sample weighting scheme for the fine-tuning data* solely based on the pre-trained model's losses. Specifically, we upweight the easy samples on which the pre-trained model's loss is low and vice versa to limit the drift from the pre-trained model. Our approach is orthogonal and yet complementary to existing methods; while such methods mostly operate on parameter or gradient space, we concentrate on the sample space. We theoretically analyze the impact of fine-tuning with our method in a linear setting, showing that it stalls learning in a certain subspace, which inhibits overfitting to the target task. We empirically demonstrate the efficacy of our method on both language and vision tasks. As an example, when fine-tuning Gemma 2 2B on MetaMathQA, our method results in only a $0.8\%$ drop in accuracy on GSM8K (another math dataset) compared to standard fine-tuning, while preserving $5.4\%$ more accuracy on the pre-training datasets.

## 1. Introduction

In the modern era of large-scale machine learning, one of the central goals is to design models capable of performing multiple tasks. Traditionally, this is achieved by training an appropriately large model over datasets of multiple tasks, ensuring that the model jointly learns multiple tasks at once. Unfortunately, it is not viable to repeat this process with every new additional task due to the scale of contemporary models, necessitating effective strategies that can essentially learn without full retraining. A resource-efficient convention in machine learning is to take a *pre-trained* model which is trained on some vast and diverse dataset, and *fine-tune* it on a new dataset/task. Such pre-trained models are typically large and expensive to train from scratch but perform well on a variety of tasks while offering a versatile basis for learning a new task.

Fine-tuning is a delicate process that should ideally serve multiple objectives simultaneously; we would like to use the base model and its capabilities to facilitate learning a strong model on the downstream task, and in the meantime, preserve the existing abilities of the pre-trained model. On this particular front, the major challenge in standard, unregulated fine-tuning is the *catastrophic forgetting* phenomenon. In broad terms, it describes the performance decline of the pre-trained model on previously observed data/tasks after fine-tuning on a new one. When the learning process for the downstream task interferes with the previously-learned representations beyond tolerable margins, the pre-trained model loses its prior capabilities and significantly underperforms on previously-learned tasks.

Mitigating catastrophic forgetting is an active area of research with many fundamental questions awaiting solutions. The key idea is to constrain the fine-tuning process to prevent the degeneration of the learned representations while guiding the learning of the new task to augment existing capabilities. The literature on the topic offers various approaches based on the available knowledge pertaining to the pre-training process. In fact, pre-training-specific data availability and how it is treated predominantly dictates the success of mitigating forgetting. In many real-life scenarios, however, the data and the training recipe used for generating the pre-trained model are not available (Radford et al., 2021; Touvron et al., 2023a;b; Grattafiori et al., 2024; Jiang et al., 2023). Naturally, one needs to approach the forgetting phenomenon accordingly to design realistic methods.

Therefore, we focus on the case in which we have ***no ac-***

---

[*]Equal contribution. Rudrajit was a PhD student at the University of Texas at Austin when this work was done. [1]University of Texas at Austin [2]Google Research. Correspondence to: Sunny Sanyal <sanyal.sunny@utexas.edu>, Hayden Prairie <hayden-prairie@utexas.edu>, Rudrajit Das <dasrudrajit@google.com>, Ali Kavis <kavis@austin.utexas.edu>, Sujay Sanghavi <sanghavi@mail.utexas.edu>.

*Proceedings of the 42$^{nd}$ International Conference on Machine Learning*, Vancouver, Canada. PMLR 267, 2025. Copyright 2025 by the author(s).

*cess to the pre-training-specific information* during the fine-tuning process; we call it the ***data-oblivious*** setting. The only piece of information available during fine-tuning is indeed the pre-trained model. Therefore, one needs to devise a strategy to regulate and guide the fine-tuning process to preserve the pre-trained model capabilities while learning the new task in the absence of prior knowledge. Under this challenging setting, we present an answer to the question:

*Can we design a principled method that mitigates forgetting during fine-tuning in the data-oblivious setting?*

In this paper, we propose **F**ine-tuning with Pre-trained **L**oss-**O**riented **W**eighting (**FLOW**) to mitigate catastrophic forgetting in the data-oblivious setting. Our key insight is upweighting the "easy" samples on which the pre-trained model's loss is low and vice versa. We believe that boosting the samples on which the pre-trained model performs well (i.e., has low loss) will introduce supervised bias to the gradient updates in favor of the pre-trained model. Intuitively, this will prevent the parameters from deviating too much from the initial pre-trained state, thus mitigating forgetting.

Some prior papers assign more importance to samples with *larger losses* to accelerate the training process (Loshchilov & Hutter, 2015; Shrivastava et al., 2016; Katharopoulos & Fleuret, 2017; Kawaguchi & Lu, 2020; Das et al., 2024). We follow the reciprocal reasoning; we tweak the fine-tuning process in favor of the pre-trained model by assigning *larger* weights to samples with *smaller* pre-trained loss values. We elaborate on this while stating our **contributions** next.

1. To mitigate forgetting, we propose **FLOW**, which fine-tunes the pre-trained model using a sample-wise weighted loss. Inspired by robust optimization ideas, we derive the $i^{\text{th}}$ sample's weight to be $\exp(-\ell_i/\tau)$, where $\ell_i$ is the $i^{\text{th}}$ sample's pre-trained loss and $\tau$ is a parameter which we set as median$(\ell_i)$ in practice. Thus, our method is essentially *parameter-free*.

2. We demonstrate the superiority of FLOW over relevant baselines (model averaging, $\ell_2$ regularization, LoRA, etc.) in both vision and language model experiments. For instance, ResNet-50 fine-tuned with FLOW on six image classification datasets achieves $\sim$ **17%** higher average accuracy (over pre-training and fine-tuning data) than standard fine-tuning, while also surpassing other relevant baselines (see Table 1). When fine-tuning Gemma 2 2B on math datasets, the corresponding improvement of FLOW over standard fine-tuning is $\sim$ **4%** (see Table 2).

3. We also empirically show that combining FLOW with existing methods for mitigating forgetting *improves the performance* of the base methods (see Tables 3 and 4).

4. We theoretically analyze the effect of fine-tuning with

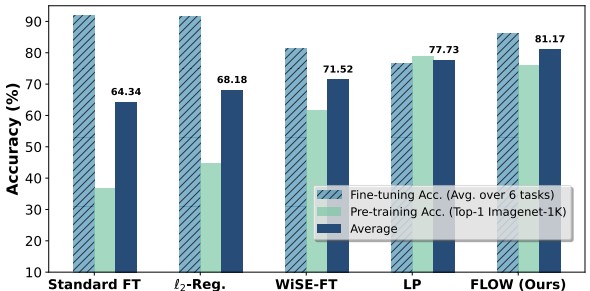

*Figure 1.* FLOW versus standard fine-tuning (FT) and relevant baselines for a ResNet-50 model pre-trained on ImageNet-1K (from Table 1). **FLOW achieves the best average accuracy** (between pre-training and target fine-tuning accuracies).

FLOW for linear models. In particular, the covariance matrix of the fine-tuning data weighted by FLOW has a small eigenvalue and *training is stalled* along the corresponding eigenvector, *impeding overfitting to the fine-tuning task* (see Remark 7.4).

We end this section with a preview of the comparison of our method FLOW with some relevant baselines (in the data-oblivious setting) in Figure 1.

## 2. Related Work

### 2.1. Mitigating Catastrophic Forgetting

We begin by summarizing the vast literature on catastrophic forgetting with a focus on prior works most relevant to our proposed setting. For a streamlined presentation, we survey prior work in two settings – data-aware and data-oblivious. Due to space limitations, we refer the readers to Appendix A for a more detailed and explanatory review of the literature.

#### 2.1.1. DATA-AWARE APPROACHES

The majority of the approaches for mitigating forgetting assume task-specific knowledge access to different extents; either (a subset of) the pre-training dataset itself or some information/statistic computed from pre-training data. Below, we describe the data-aware approaches based on how they make use of task-specific knowledge.

**Regularization-based methods.** This line of work aims to preserve existing capabilities by keeping the parameters close to the pre-trained model. The key idea is to introduce task-specific regularization to penalize modifications along the "important" directions for the old tasks (Ahn et al., 2019). Kirkpatrick et al. (2016) introduces the elastic weight consolidation (EWC) algorithm, which estimates the important directions by approximating the Fisher information matrix. Several variants of EWC have been proposed (Schwarz et al., 2018; Ritter et al., 2018; Lee et al., 2020; Liu et al., 2018).

Zenke et al. (2017); Aljundi et al. (2018) infer the importance of each parameter by their variational effect on the outputs. In a similar spirit, Lee et al. (2017) aims to match the posteriors of the pre-trained and fine-tuned models.

**Optimization-driven methods.** Another perspective to mitigating forgetting is guiding the optimization process by constraining the algorithms directly as opposed to manipulating the loss function. The core idea is to keep track of "important directions" for the old tasks, and train on the new task "orthogonally." This could be done by storing prior data samples or gradients in a buffer (Lopez-Paz & Ranzato, 2017; Farajtabar et al., 2020; Chaudhry et al., 2019a) or by incrementally expanding the subspace of important directions without storing task-specific information (Zeng et al., 2019; Wang et al., 2021; 2023b).

**Replay-based methods.** A more direct approach is to store old task samples in buffers and introduce them into the training process for the new task to refresh task-specific representations periodically. There are several components to such methods. Some prior work focus on *data selection* based on the nature of old data access (Rebuffi et al., 2017; Aljundi et al., 2019; Bang et al., 2021; Chaudhry et al., 2019b; Isele & Cosgun, 2018; De Lange & Tuytelaars, 2021; Borsos et al., 2020; Tiwari et al., 2021) (e.g., streaming versus on-demand). Another important perspective is the *re-introduction strategy* of the stored information into the fine-tuning process (Silver & Mercer, 2002; Li & Hoiem, 2016; Triki et al., 2017; Lee et al., 2019b; Dhar et al., 2019; Rebuffi et al., 2017; Riemer et al., 2019; Chaudhry et al., 2019b; De Lange & Tuytelaars, 2021; Tiwari et al., 2021).

**Architecture-driven methods.** Another technique to limit interference between tasks is to allocate a separate trainable set of parameters per task. This could be done by initializing a sub-networks per new task (Rusu et al., 2016; Aljundi et al., 2017; Collier et al., 2020; Rajasegaran et al., 2019; Ramesh & Chaudhari, 2021; Wang et al., 2023a; 2022a), gradually expanding the parameters of a base network (Yoon et al., 2018; Ostapenko et al., 2019; Hung et al., 2019), or segregating a fixed model into task-specific subsets (Mallya et al., 2018; Kang et al., 2022; Serra et al., 2018; Wortsman et al., 2020; Mallya & Lazebnik, 2017; Mustafa B Gurbuz, 2022; Jung et al., 2020). The main downside with this line of work is that task identities must be known for inference to (de)activate relevant sub-networks (Aljundi et al., 2017).

### 2.1.2. DATA-OBLIVIOUS APPROACHES

In the less-explored data-oblivious setting, it is particularly challenging to devise a principled approach, as there is no access to any data-specific information, except for the pre-trained model. One line of work explores the simple idea of "model averaging" (MA) which essentially does a convex combination of the parameters of the pre-trained

model and that of the fully fine-tuned model for the new task. MA and more sophisticated model merging variants have been studied in relevant context to forgetting (Lubana et al., 2021; Wortsman et al., 2021; Ilharco et al., 2023; Lin et al., 2023; Kleiman et al., 2025). Some recent works Chen et al. (2024b); Panda et al. (2024) introduce different strategies to selectively update a subset of parameters in a pre-training data-agnostic manner. Finally, Biderman et al. (2024) has shown that LoRA (Hu et al., 2022) could be effective for mitigating catastrophic forgetting in transformers. Unlike the methods discussed above which focus on the parameter or gradient space, ours focuses on the sample space.

### 2.2. Sample Selection and Weighting

Sample-wise importance selection/weighting has been studied in optimization papers (Needell et al., 2014; Zhao & Zhang, 2015; Alain et al., 2015; Stich et al., 2017) and ML papers (Loshchilov & Hutter, 2015; Shrivastava et al., 2016; Katharopoulos & Fleuret, 2017; 2018; Kawaguchi & Lu, 2020; Das et al., 2024) to speed up the optimization/training process by reducing the variance of the gradient updates. Such papers advocate focusing on "hard" samples with high-gradient norms or losses. In contrast, we focus on "easy" samples to mitigate forgetting. Another line of work focuses on robust learning under uncertain data distributions. Distributionally robust optimization (DRO) proposes to minimize the worst-case weighted loss, where the sample weights are constrained or regularized (Ben-Tal et al., 2013; Levy et al., 2020; Duchi & Namkoong, 2021; Qi et al., 2021). Some recent works (Xie et al., 2024; Chen et al., 2024a; Anonymous, 2025) propose dynamic sample-weighting strategies for LLM training based on the previously discussed ideas.

## 3. Notation and Definitions

$\mathbb{1}(.)$ denotes the indicator variable. For any $n \in \mathbb{N}$, the set $\{1, \dots, n\}$ is denoted by $[n]$. Vectors and matrices are in lowercase and uppercase bold font, respectively. The $\ell_p$ norm of a vector $\mathbf{v}$ is denoted by $\|\mathbf{v}\|_p$. The inner product between two vectors $\mathbf{v}$ and $\mathbf{v}'$ is denoted as $\langle \mathbf{v}, \mathbf{v}' \rangle$. A set of $n$ linearly independent $n$-dimensional vectors $\{\mathbf{u}_1, \dots, \mathbf{u}_n\}$ is said to be an orthonormal basis for $\mathbb{R}^n$ if $\langle \mathbf{u}_i, \mathbf{u}_j \rangle = \mathbb{1}(i = j)$. A vector $\mathbf{v} = [\mathrm{v}_1, \dots, \mathrm{v}_n]^\top$ is said to belong to the $n$-dimensional probability simplex $\boldsymbol{\Delta}_n$ if $\sum_{i=1}^n \mathrm{v}_i = 1$ and $\mathrm{v}_i \geq 0 \, \forall \, i \in [n]$. For any $n \in \mathbb{N}$, $\mathbf{I}_n$ denotes the identity matrix of dimension $n$.

## 4. Proposed Algorithm

Our proposed algorithm consists of two main steps: *(i)* computing weights for the samples based on their respective pre-trained loss values; and *(ii)* fine-tuning with a weighted loss wherein the per-sample losses are scaled by their respec-

tive weights. The sample-wise weights are computed once and used throughout the entire fine-tuning process. We formally state our proposed fine-tuning protocol in Algorithm 1 and delve into its design details in the sequel.

---

**Algorithm 1** Fine-tuning with Pre-trained **L**oss-**O**riented **W**eighting (**FLOW**)

---

**Input:** Pre-trained model $\boldsymbol{\theta}^*$, dataset $\{(\mathbf{x}_i, \mathbf{y}_i)\}_{i=1}^n$ for the new task, and temperature parameter $\tau$.

$f_i(\boldsymbol{\theta}) \to i^{\text{th}}$ sample's loss at $\boldsymbol{\theta}$, with a non-negative loss function (e.g., cross-entropy loss).

1. Compute sample weights: $w_i = \exp\left(-\frac{f_i(\boldsymbol{\theta}^*)}{\tau}\right)$.

2. Weighted loss: $\mathcal{L}(\boldsymbol{\theta}) = \sum_{i=1}^n w_i f_i(\boldsymbol{\theta})$.

3. Fine-tune with weighted loss: $\widehat{\boldsymbol{\theta}}^* := \arg\min_{\boldsymbol{\theta}} \mathcal{L}(\boldsymbol{\theta})$.

**Output:** Fine-tuned model $\widehat{\boldsymbol{\theta}}^*$.

---

**Remark 4.1.** *Depending on the setting, our model might have task-specific components, such as per-task prediction heads (e.g., in vision). Algorithm 1 can be slightly modified in the presence of task-specific components to enhance performance. Refer to Appendix B for these modifications.*

**Remark 4.2.** *As a heuristic prescription, we set* $\tau = $ median $(f_i(\boldsymbol{\theta}^*))$ *in all our experiments (unless otherwise stated), which leads to consistently good performance. **Thus, our algorithm is essentially parameter-free in practice**.*

**Algorithm design.** Our main intuition is that we can control forgetting by not drifting away too much from the pre-trained model (i.e., $\boldsymbol{\theta}^*$) during fine-tuning. In the presence of pre-training data, this is done by introducing data-dependent constraints on the parameter space or gradient space. Since we have *no access* to pre-training data, we redirect our focus towards *strategies on the **sample space** depending only on the **pre-trained model***.

To that end, we propose to infer the easiness of each sample of the fine-tuning dataset with respect to the pre-trained model, based on the per-sample losses $f_i(\boldsymbol{\theta}^*)$'s (see Alg. 1). We say that the $i^{\text{th}}$ sample is "easy" if $f_i(\boldsymbol{\theta}^*)$ is "small".[1] Intuitively, prioritizing the "easy" samples during fine-tuning would limit the drift from $\boldsymbol{\theta}^*$. On the other hand, over-focusing on the "easy" samples would probably lead to poor performance on the fine-tuning task. Thus, it is important to strike a balance.

Let us formalize these ideas mathematically. For fine-tuning on the new task, let us consider the objective function $\mathcal{L}_{\boldsymbol{\pi}}(\boldsymbol{\theta}) = \sum_{i=1}^n \pi_i f_i(\boldsymbol{\theta})$, where $\boldsymbol{\pi} = [\pi_1, \ldots, \pi_n]^\top$ is a **static** design-choice $\in \boldsymbol{\Delta}_n$ (i.e., $\sum_{i=1}^n \pi_i = 1$ and $\pi_i \geq 0$ $\forall i \in [n]$) which we allow to only depend on the pre-trained

---

[1]This is not a formal definition and so "small" is not quantified.

---

model's losses $\{f_i(\boldsymbol{\theta}^*)\}_{i=1}^n$ (and *not* the current model's losses $\{f_i(\boldsymbol{\theta})\}_{i=1}^n$). We would like to design $\boldsymbol{\pi}$ so that:

1. for all $i \neq j$ such that $f_i(\boldsymbol{\theta}^*) \leq f_j(\boldsymbol{\theta}^*)$, $\pi_i \geq \pi_j$,

2. $\boldsymbol{\pi}$ does not concentrate around one or a few samples but rather spreads uniformly over the samples.

These two requirements can be enforced by minimizing the following function (w.r.t. $\boldsymbol{\pi}$) involving negative *entropic regularization*:

$$g(\boldsymbol{\pi}) = \sum_{i=1}^n \pi_i f_i(\boldsymbol{\theta}^*) + \tau \sum_{i=1}^n \pi_i \log \pi_i. \quad (1)$$

Here $\tau > 0$ is a parameter controlling the extent of the second requirement which is facilitated by the entropy term. We now state the minimizer of $g(\boldsymbol{\pi})$ (proof is in Appendix C).

**Proposition 4.3.** *Let* $\boldsymbol{\pi}^* = [\pi_1^*, \ldots, \pi_n^*]^\top = \arg\min_{\boldsymbol{\pi} \in \boldsymbol{\Delta}_n} g(\boldsymbol{\pi})$. *Then we have* $\pi_i^* = \frac{1}{Z} \exp\left(-\frac{f_i(\boldsymbol{\theta}^*)}{\tau}\right)$, *where $Z$ is the normalizing factor.*

Modulo the normalizing factor $Z$ (it does not matter when optimizing w.r.t. $\boldsymbol{\theta}$), note that $w_i$ and $\mathcal{L}(\boldsymbol{\theta})$ in Algorithm 1 are equivalent to $\pi_i^*$ and $\mathcal{L}_{\boldsymbol{\pi}^*}(\boldsymbol{\theta})$, respectively.

**Distributionally robust optimization (DRO) perspective.** Our formulation above is motivated by prior work on DRO (Qi et al., 2021), but it is **exactly the opposite** of DRO in spirit. Specifically, in our setting, Qi et al. (2021) consider the following min-max problem:

$$\min_{\boldsymbol{\theta}} \max_{\boldsymbol{\pi} \in \boldsymbol{\Delta}_n} \sum_{i=1}^n \pi_i f_i(\boldsymbol{\theta}) - \tau \sum_{i=1}^n \pi_i \log \pi_i. \quad (2)$$

The first term in Eq. (2) is the *worst-case* weighted loss at $\boldsymbol{\theta}$, while the second term (i.e., entropic regularization) promotes uniform weights. The optimal solution to the inner max function w.r.t. $\boldsymbol{\pi}$ turns out to be $\pi_i^* \propto \exp\left(\frac{f_i(\boldsymbol{\theta})}{\tau}\right)$. Note that this is essentially the *inverse of our weighting function* (modulo the normalizing factor) because it assigns a higher weight to samples with larger losses (i.e., the "hard" samples). The weighting function of DRO would be very conducive to forgetting because it focuses more on the "hard" samples. Further, our weighting function is static (or one-shot) as it depends only on the losses at $\boldsymbol{\theta}^*$. On the other hand, the weighting function of DRO is dynamic (i.e., it depends on the current point $\boldsymbol{\theta}$). In fact, after plugging in the optimal value of $\pi$ into Eq. (2) and simplifying, the DRO objective reduces to $\min_{\boldsymbol{\theta}} \sum_{i=1}^n \exp\left(\frac{f_i(\boldsymbol{\theta})}{\tau}\right)$; this is noticeably different from our objective $\mathcal{L}(\boldsymbol{\theta})$ in Algorithm 1.

## 5. Experimental Setup

We empirically evaluate the performance of FLOW (Algorithm 1) on vision and language tasks, showcasing its effectiveness across different model architectures and modalities.[2] Here, we explain details of our experiments: baselines, model architectures, datasets, and evaluation metrics.

**Baselines.** In our language and vision experiments, we compare FLOW against relevant baselines in the *data-oblivious setting*, namely, standard fine-tuning (fine-tuning with vanilla unweighted loss), $\ell_2$-regularization [following Kirkpatrick et al. (2016)], and WiSE-FT (Wortsman et al., 2021) (model averaging of pre-trained and standard fine-tuned models). Additionally, we compare against linear probing (fine-tuning only the classification head, keeping the body frozen) and low-rank adaptation (LoRA) (Hu et al., 2022) in language experiments. More details on the baselines can be found in Appendix G.1.

### 5.1. Vision Experiments

We compare the performance of FLOW and associated baselines in a transfer learning setup.

**Models.** We experimented with ResNet-18 and ResNet-50 (Wightman et al., workshop) pre-trained on Imagenet-1K (IN-1K).

**Datasets.** We used seven widely-used image classification datasets: CIFAR-10 (Krizhevsky, 2009), CIFAR-100 (Krizhevsky, 2009), Flowers102 (Nilsback & Zisserman, 2008), Caltech101 (Li et al., 2022), Cars (Krause et al., 2013), and Dogs (Parkhi et al., 2012).

**Evaluation metrics.** Vision models are trained with task-specific parts, such as classification head (head) and batch-norm (BN); see Appendix B for how FLOW works with with task-specific parts. Forgetting is measured by how much the model's top-1 validation accuracy on ImageNet-1K (subsequently referred to as IN-1K accuracy) reduces after fine-tuning. We report the fine-tuning performance in terms of *average fine-tuning accuracy* over all the related datasets following (Goyal et al., 2023; Ilharco et al., 2023). For IN-1K evaluation *after* fine-tuning, we replace the task-specific components of the fine-tuned model with their pre-trained counterparts. An extended discussion on experimental details, evaluation, and hyper-parameters are in Appendix G.4. We also report the **average** of IN-1K accuracy and averaged fine-tuning accuracy for each method; this is a reasonable *unified metric* to evaluate the performance of a method jointly on the pre-training and fine-tuning data.

---

[2]Our code is publicly available `here`.

### 5.2. Language Model Experiments

We follow a similar setup to Biderman et al. (2024); Chen et al. (2024b), where a language model's general capabilities are evaluated before and after fine-tuning on a mathematical reasoning dataset. All training for language experiments is done with HuggingFace `peft` (Mangrulkar et al., 2022), `transformers` (Wolf et al., 2020), `datasets` (Lhoest et al., 2021), and `accelerate` (Gugger et al., 2022).

**Models.** We use Gemma 2 2B (Team et al., 2024) and Llama 3.2 3B (Grattafiori et al., 2024) as our base language models. Further details on training hyper-parameters can be found in Appendix G.2.

**Datasets.** Following previous work (Biderman et al., 2024; Chen et al., 2024b), we fine-tune on MetaMathQA (Yu et al., 2023), a mathematical reasoning dataset that is bootstrapped from the training set of GSM8K (Cobbe et al., 2021) and MATH (Hendrycks et al., 2021b) using a LLM. We train with all 395K samples in MetaMathQA.

**Evaluation metrics.** To evaluate the validity of FLOW, we break down our metrics into *general capability* and *target fine-tuning* evaluations. To evaluate general capabilities, we again follow a similar setup to Chen et al. (2024b), where we use commonsense reasoning, 5-shot MMLU (Hendrycks et al., 2021a), and 3-shot MBPP (Austin et al., 2021) metrics. To evaluate the target domain, we use 5-shot GSM8K (Cobbe et al., 2021). All evaluations are performed with `lm-evaluation-harness` (Gao et al., 2024). More details on evaluation and the commonsense metric can be found in Appendix G.3. Similar to vision, we also report the **average** of general capabilities and the target fine-tuning accuracies as a *unified metric*.

## 6. Experimental Results

### 6.1. Comparing FLOW and Related Baselines

For our **vision** experiments on ResNets, Table 1 lists the accuracies of all the baselines and FLOW. We observe similar trends for both ResNet models, so we discuss the results for the larger ResNet-50 model here. The pre-trained ResNet-50 model achieves a top-1 accuracy of 79.02% on ImageNet-1K's validation set. Standard fine-tuning experiences a significant 42.11% drop in IN-1K accuracy, while achieving an average fine-tuning accuracy of 91.78% across the target datasets. In contrast, FLOW suffers only a 2.93% drop in IN-1K accuracy and exhibits a reasonable 86.25% average accuracy on target fine-tuning datasets, demonstrating a significant improvement over standard fine-tuning. Overall, FLOW's **average** on IN-1K and target domain accuracy is **16.83% higher** than standard fine-tuning.

Going beyond standard fine-tuning, our results in Table 1 show that FLOW comprehensively outperforms other base-

*Table 1.* **Performance of FLOW with ResNet vision models.** **Bolded** and underlined values indicate the **best** and second-best accuracies within each column (and for each model). Deltas (in color) for IN-1K and target performance are computed w.r.t. the pre-trained and standard fine-tuned models. **FLOW attains the best average accuracy** and is better than the second-best method (linear probing) by 2.94% and 3.44% for ResNet-18 and ResNet-50, respectively.

| | Method | IN-1K Acc. | Target Acc. | Average |
|---|---|---|---|---|
| **ResNet-18** | Pre-trained | **69.76** (+0.00) | – | – |
| | Standard FT | 19.58 (-50.18) | **89.07** (+0.00) | 54.60 |
| | Linear Probe | **69.76** (+0.00) | 73.57 (-15.50) | 71.63 |
| | $\ell_2$-Reg. | 34.78 (-34.98) | 88.12 (-0.95) | 61.45 |
| | WiSE-FT | 54.15 (-15.61) | 80.23 (-8.84) | 67.19 |
| | **FLOW** (Ours) | 65.21 (-4.55) | 83.93 (-5.14) | **74.57** |
| **ResNet-50** | Pre-trained | **79.02** (+0.00) | – | – |
| | Standard FT | 36.91 (-42.11) | **91.78** (+0.00) | 64.34 |
| | Linear Probe | **79.02** (+0.00) | 76.45 (-15.33) | 77.73 |
| | $\ell_2$-Reg. | 44.78 (-34.24) | 91.58 (-0.20) | 68.18 |
| | WiSE-FT | 61.65 (-17.37) | 81.38 (-10.40) | 71.52 |
| | **FLOW** (Ours) | 76.09 (-2.93) | 86.25 (-5.53) | **81.17** |

lines. Interestingly, despite its simplicity, linear probing is the second-best method. Overall, FLOW outperforms other baselines, when **averaging** IN-1K and target fine-tuning accuracies, by **a 3.44% advantage** over the closest competitor, linear probing. Although linear probing completely prevents forgetting, it learns significantly less during fine-tuning compared to FLOW. The accuracies on individual fine-tuning datasets and corresponding accuracies for IN-1K can be found in Appendix H.

Additionally, in Appendix H.1, we compare FLOW with a distillation-based approach for mitigating forgetting called "learning without forgetting" (LwF) (Li & Hoiem, 2016) on ViT-B/16, which is a larger model than ResNets and on the Food101 (Bossard et al., 2014) dataset, which is relatively larger than each of the six datasets here. In summary, FLOW outperforms LwF in terms of average accuracy; while LwF has the edge on fine-tuning performance, FLOW does better on the forgetting front.

Our **language model** results are in Table 2. Results for Gemma 2 2B show that FLOW helps preserve (and even somewhat enhance) the general capabilities of the pre-trained model. Specifically, compared to standard fine-tuning, FLOW improves general capability accuracy by 2.52% in commonsense reasoning, 3.73% in MMLU, and 10.00% in MBPP, with a minor degradation of 0.83% in GSM8K. We see a similar trend in our Llama 3.2 3B experiments. Furthermore, while alternative baselines show specific strengths (such as WiSE-FT's general capability performance and $\ell_2$-regularization's target fine-tuning performance), **FLOW outperforms all baselines, on average,** for both models, striking the best balance between preserving

general capabilities and achieving good target fine-tuning performance. Additional details on commonsense reasoning results are in Appendix I.1 and an ablation for our choice of sample weighting in LLMs is in Appendix I.2.

In summary, *FLOW strikes a good balance between learning a new task and retaining knowledge from pre-training.*

### 6.2. Combining FLOW with Baselines

To complement our results in Tables 1 and 2, we investigate the performance of baselines when **combined with FLOW**. In the vision setting, we consider uniform model averaging with WiSE-FT (with $\alpha = 0.5$) and report its performance with and without FLOW in Table 3. Interestingly, averaging the pre-trained IN-1K model and the fine-tuned model obtained with FLOW **improves** over standard WiSE-FT (i.e., averaging the pre-trained IN-1K model and the standard fine-tuned model) by **4.18%** and **4.52%** for ResNet-18 and ResNet-50, respectively, in average performance.

Further, as seen in Table 4, FLOW **boosts the performance** of other baselines in language modeling. When combined with $\ell_2$-regularization, we observe improvements in general capability between 0.5% and 1.80%, with only a 0.83% reduction in GSM8K performance. Furthermore, the integration of FLOW with LoRA yields even stronger results, enhancing general capability performance by 1.07% to 3.40%, while simultaneously improving GSM8K performance by 1.06%. Further details and discussion combining FLOW with $\ell_2$-regularization and LoRA are in Appendix I.1.

## 7. Theoretical Analysis

Here we consider linear pre-training and fine-tuning tasks[3] and theoretically analyze the effect of fine-tuning with our proposed method FLOW (Alg. 1). Specifically, we compare the non-asymptotic trajectories of FLOW and vanilla fine-tuning. A key insight of our analysis is that FLOW stalls training in a certain direction, impeding overfitting to the fine-tuning task (see Remark 7.4). We also demonstrate that FLOW goes beyond the simple idea of model averaging (see Remark 7.5).

We begin by describing the problem setting.

**Pre-training task:** The label $y \in \mathbb{R}$ for a $d$-dimensional data point $\mathbf{x} \sim \mathcal{P}$ is given by $y = \langle \boldsymbol{\theta}_*, \mathbf{x} \rangle$, where $\boldsymbol{\theta}_* \in \mathbb{R}^d$ is the ground-truth model. Let $\mathcal{D}$ denote the joint distribution of $(\mathbf{x}, y)$, where $\mathbf{x} \sim \mathcal{P}$. Let $\boldsymbol{\Sigma} = \mathbb{E}_{\mathbf{x} \sim \mathcal{P}}[\mathbf{x}\mathbf{x}^\top]$ be the data covariance matrix. Without loss of generality, let $\boldsymbol{\Sigma} \succeq \mathbf{I}_d$.

**Fine-tuning task:** The label $\widetilde{y} \in \mathbb{R}$ for a $d$-dimensional data point $\widetilde{\mathbf{x}} \sim \widetilde{\mathcal{P}}$ is given by $\widetilde{y} = \langle \widetilde{\boldsymbol{\theta}}_*, \widetilde{\mathbf{x}} \rangle$, where $\widetilde{\boldsymbol{\theta}}_* \in \mathbb{R}^d$ is the

---

[3]Our insights carry over to neural networks following the dynamics of linear models under gradient descent (Lee et al., 2019a).

*Table 2.* **Performance of FLOW with LLMs.** After fine-tuning Gemma 2 2B and Llama 3.2 3B on MetaMathQA, we compare the target fine-tuning performance (GSM8K) with general capability performance. **Bolded** and underlined values indicate the **best** and second-best results within each column (and for each model). Deltas (in color) for general capability metrics and fine-tuning metrics are computed w.r.t. the pre-trained and standard fine-tuned model's accuracy, respectively. We see that **FLOW, on average, has the best performance** on general capabilities and target domain, achieving within $\sim 0.8\%$ (Gemma 2 2B) and $\sim 1.4\%$ (Llama 3.2 3B) of standard fine-tuning's target performance, while significantly mitigating the degradation of general pre-training capabilities in comparison to other baselines.

| | Method | General Capability Acc. | | | Target Acc. | |
|---|---|---|---|---|---|---|
| | | **Commonsense** | **MMLU** | **MBPP** | **GSM8K** | **Average** |
| **Gemma 2 2B** | Pre-trained | 57.23 (+0.00) | 49.59 (+0.00) | **28.40** (+0.00) | 24.49 (-38.89) | 40.79 |
| | Standard Fine-tuning | 55.07 (-2.16) | 45.59 (-4.00) | 16.80 (-11.60) | **63.38** (+0.00) | 46.31 |
| | WiSE-FT ($\alpha = 0.5$) | 57.28 (+0.05) | **50.13** (+0.54) | 25.60 (-2.80) | 53.30 (-10.08) | 47.60 |
| | LoRA ($r = 64$) | 55.67 (-1.56) | 44.28 (-5.31) | 25.80 (-2.60) | 60.43 (-2.95) | 47.05 |
| | $\ell_2$-Regularization | 57.01 (-0.22) | 48.43 (-1.16) | 24.80 (-3.60) | 62.85 (-0.53) | 49.19 |
| | **FLOW** (Ours) | **57.59** (+0.36) | 49.31 (-0.28) | 26.80 (-1.60) | 62.55 (-0.83) | **49.98** |
| **Llama 3.2 3B** | Pre-trained | 54.48 (+0.00) | **54.34** (+0.00) | **38.00** (+0.00) | 26.01 (-40.94) | 44.28 |
| | Standard Fine-tuning | 50.68 (-3.80) | 45.29 (-9.05) | 17.80 (-20.20) | **66.95** (+0.00) | 46.10 |
| | WiSE-FT ($\alpha = 0.5$) | **54.54** (+0.04) | 53.33 (-1.01) | 34.60 (-3.40) | 57.01 (-9.94) | 50.75 |
| | LoRA ($r = 64$) | 53.10 (-1.38) | 50.95 (-3.39) | 34.00 (-4.00) | 63.84 (-3.15) | 51.66 |
| | $\ell_2$-Regularization | 53.60 (-0.88) | 51.28 (-3.06) | 33.60 (-4.40) | 66.87 (-0.08) | 52.30 |
| | **FLOW** (Ours) | 54.30 (-0.18) | 51.86 (-2.48) | 36.00 (-2.00) | 65.58 (-1.37) | **52.87** |

*Table 3.* **WiSE-FT with FLOW vs. WiSE-FT in vision.** "WiSE-FT+" denotes WiSE-FT with FLOW in the table. Comparison here is in the same setting as Table 1. Note that **WiSE-FT+ is significantly better than WiSE-FT**.

| | Method | IN-1K | Target | Average |
|---|---|---|---|---|
| **ResNet-18** | WiSE-FT | 54.15 | 80.23 | 67.19 |
| | WiSE-FT+ | 68.71 | 74.03 | **71.37** |
| **ResNet-50** | WiSE-FT | 61.65 | 81.38 | 71.52 |
| | WiSE-FT+ | 78.29 | 73.80 | **76.04** |

*Table 4.* **$\ell_2$-Reg./LoRA + FLOW vs. $\ell_2$-Reg./LoRA in language.** "$\ell_2$+" and "LoRA+" denote $\ell_2$-Reg. with FLOW and LoRA with FLOW, respectively. The results below are for Gemma 2 2B in the same setup as Table 2. We let A1, A2, A3, and B1 represent Commonsense, MMLU, MBPP, and GSM8K, respectively. We see that **$\ell_2$+ and LoRA+ are better than $\ell_2$ and LoRA**, respectively.

| Method | A1 | A2 | A3 | B1 | Avg. |
|---|---|---|---|---|---|
| $\ell_2$ | 57.01 | 48.43 | 24.80 | **62.85** | 49.19 |
| $\ell_2$+ | **57.53** | **49.38** | **26.60** | 62.02 | **49.79** |
| LoRA | 55.67 | 44.28 | 25.80 | 60.43 | 47.05 |
| LoRA+ | **56.74** | **47.68** | **28.80** | **61.49** | **49.31** |

ground-truth model. Let $\widetilde{\mathcal{D}}$ denote the joint distribution of $(\widetilde{\mathbf{x}}, \widetilde{\mathbf{y}})$, where $\widetilde{\mathbf{x}} \sim \widetilde{\mathcal{P}}$. Also, let

$$\mathbf{e} := \boldsymbol{\theta}_* - \widetilde{\boldsymbol{\theta}}_*, \overline{\mathbf{e}} := \mathbf{e}/\|\mathbf{e}\|_2,$$

and $\overline{\mathbf{e}}_\perp$ be a unit vector orthogonal to $\overline{\mathbf{e}}$. We consider the case of $\widetilde{\mathcal{P}} = \mathcal{N}(\vec{\mathbf{0}}_d, \widetilde{\boldsymbol{\Sigma}})$, where

$$\widetilde{\boldsymbol{\Sigma}} = \mathbf{I}_d + \rho\big(\overline{\mathbf{e}}\overline{\mathbf{e}}_\perp^\top + \overline{\mathbf{e}}_\perp\overline{\mathbf{e}}^\top\big), \tag{3}$$

where $\rho \in [0, 1)$ is a constant. Note that $\widetilde{\boldsymbol{\Sigma}}$ is the data covariance matrix here.

**Remark 7.1 (Regarding $\widetilde{\boldsymbol{\Sigma}}$).** *We study the case of $\widetilde{\boldsymbol{\Sigma}}$ as given in Eq. (3) because it is the minimal analytically tractable case where we can show that FLOW goes beyond model averaging (MA) (see Remark 7.5). Specifically, if $\rho = 0$ and $\widetilde{\boldsymbol{\Sigma}} = \mathbf{I}_d$, then FLOW reduces to MA. Moreover, for an arbitrary $\widetilde{\boldsymbol{\Sigma}}$, characterizing the eigen-spectrum of the matrix dictating the trajectory of FLOW becomes intractable. For the analysis to be tractable, we need some*

*relationship between $\widetilde{\boldsymbol{\Sigma}}$ and $\mathbf{e}$ (i.e., the difference between the optima of the pre-training and fine-tuning tasks).*[4]

For a model parameterized by $\boldsymbol{\theta} \in \mathbb{R}^d$, let

$$\mathrm{err}_1(\boldsymbol{\theta}) := \mathbb{E}_{\mathcal{D}}\Big[\big(y - \langle\boldsymbol{\theta}, \mathbf{x}\rangle\big)^2\Big] = \big(\boldsymbol{\theta} - \boldsymbol{\theta}_*\big)^\top \boldsymbol{\Sigma} \big(\boldsymbol{\theta} - \boldsymbol{\theta}_*\big),$$

$$\mathrm{err}_2(\boldsymbol{\theta}) := \mathbb{E}_{\widetilde{\mathcal{D}}}\Big[\big(\widetilde{y} - \langle\boldsymbol{\theta}, \widetilde{\mathbf{x}}\rangle\big)^2\Big] = \big(\boldsymbol{\theta} - \widetilde{\boldsymbol{\theta}}_*\big)^\top \widetilde{\boldsymbol{\Sigma}} \big(\boldsymbol{\theta} - \widetilde{\boldsymbol{\theta}}_*\big) \tag{4}$$

be the population errors on the pre-training and fine-tuning tasks, respectively. Also, the total error with $\boldsymbol{\theta}$ on the two tasks is denoted by $\mathrm{err}_{\mathrm{tot}}(\boldsymbol{\theta}) = \mathrm{err}_1(\boldsymbol{\theta}) + \mathrm{err}_2(\boldsymbol{\theta})$.

We assume that initially, we learn $\boldsymbol{\theta}_*$ with the pre-training data; so $\boldsymbol{\theta}_*$ is our pre-trained model. Note that

$$\mathrm{err}_{\mathrm{tot}}(\boldsymbol{\theta}_*) = \mathrm{err}_2(\boldsymbol{\theta}_*) = \mathbf{e}^\top \widetilde{\boldsymbol{\Sigma}} \mathbf{e} = \|\mathbf{e}\|_2^2, \tag{5}$$

where the last step follows by using $\widetilde{\boldsymbol{\Sigma}}$ from Equation (3).

---

[4]See Appendix E for more details.

We start fine-tuning starting from $\boldsymbol{\theta}_*$. Specifically, we assume access to the population $(\widetilde{\mathbf{x}}, \widetilde{y}) \sim \widetilde{\mathcal{D}}$ of the fine-tuning task, but we lose access to the pre-training data.

**Vanilla fine-tuning (FT):** We minimize $\mathrm{err}_2(\boldsymbol{\theta})$ (Eq. (4)) with gradient descent (GD) starting from $\boldsymbol{\theta}_*$ using a constant learning rate $\overline{\eta}$. Our iterate $\overline{\boldsymbol{\theta}}_K$ at the $K^{\text{th}}$ iteration is given by (using the value of $\widetilde{\boldsymbol{\Sigma}}$ from Eq. (3) and $\boldsymbol{\theta}_* - \widetilde{\boldsymbol{\theta}}_* = \mathbf{e}$):

$$\overline{\boldsymbol{\theta}}_K = \widetilde{\boldsymbol{\theta}}_* + \left(\mathbf{I}_d - 2\overline{\eta}\left(\mathbf{I}_d + \rho\big(\overline{\mathbf{e}}\overline{\mathbf{e}}_\perp^\top + \overline{\mathbf{e}}_\perp\overline{\mathbf{e}}^\top\big)\right)\right)^K \mathbf{e}. \quad (6)$$

**FLOW:** For some temperature $\tau$, the weight of $(\widetilde{\mathbf{x}}, \widetilde{y}) \sim \widetilde{\mathcal{D}}$ is $w(\widetilde{\mathbf{x}}, \widetilde{y}) = \exp\left(-\frac{(\widetilde{y} - \langle\boldsymbol{\theta}_*, \widetilde{\mathbf{x}}\rangle)^2}{\tau}\right)$. We minimize

$$\widehat{\mathrm{err}}_2(\widehat{\boldsymbol{\theta}}) := \mathbb{E}_{\widetilde{\mathcal{D}}}\left[w(\widetilde{\mathbf{x}}, \widetilde{y})\big(\widetilde{y} - \langle\widehat{\boldsymbol{\theta}}, \widetilde{\mathbf{x}}\rangle\big)^2\right], \quad (7)$$

with GD starting from $\boldsymbol{\theta}_*$ using a constant learning rate $\widehat{\eta}$. Suppose our iterate at the $K^{\text{th}}$ iteration is $\widehat{\boldsymbol{\theta}}_K$.

**Theorem 7.2 (FLOW).** *Let* $\mu = \left(\frac{\tau}{\tau + 2\|\mathbf{e}\|_2^2}\right)^{1/2}$. *Then:*

$$\widehat{\boldsymbol{\theta}}_K = \widetilde{\boldsymbol{\theta}}_* + \left(\mathbf{I}_d - 2\widehat{\eta}\widetilde{\boldsymbol{\Sigma}}'\right)^K \mathbf{e}, \quad (8)$$

*where* $\widetilde{\boldsymbol{\Sigma}}' = \mu\big(\mathbf{I}_d - \mathbf{Q}\big)$ *with*

$$\mathbf{Q} = (1 - \mu^2)\overline{\mathbf{e}}\overline{\mathbf{e}}^\top + \rho^2(1 - \mu^2)\overline{\mathbf{e}}_\perp\overline{\mathbf{e}}_\perp^\top$$
$$- \rho\mu^2\big(\overline{\mathbf{e}}\overline{\mathbf{e}}_\perp^\top + \overline{\mathbf{e}}_\perp\overline{\mathbf{e}}^\top\big). \quad (9)$$

We prove Thm. 7.2 in Appendix D. The **main technical challenge** is the evaluation of $\widetilde{\boldsymbol{\Sigma}}'$, viz., the covariance matrix of the *weighted* fine-tuning data; see Lemma F.1 for this.

Now, we are going to compare vanilla FT (6) with $\overline{\eta} = \frac{1}{2}$ and FLOW (8) with $\widehat{\eta} = \frac{1}{2\mu}$. We believe these are comparable learning rates for vanilla FT and FLOW because the resultant matrices (Eqs. (10) and (11)) dictating the convergence of both methods have exactly two non-zero eigenvalues and the corresponding eigenvectors lie in the span of $\overline{\mathbf{e}}$ and $\overline{\mathbf{e}}_\perp$. Plugging in $\overline{\eta} = \frac{1}{2}$ into Eq. (6), we get:

$$\overline{\boldsymbol{\theta}}_K = \widetilde{\boldsymbol{\theta}}_* + \mathbf{P}^K\mathbf{e}, \text{ with } \mathbf{P} = -\rho\big(\overline{\mathbf{e}}\overline{\mathbf{e}}_\perp^\top + \overline{\mathbf{e}}_\perp\overline{\mathbf{e}}^\top\big) \quad (10)$$

for vanilla FT. Plugging in $\widehat{\eta} = \frac{1}{2\mu}$ into Eq. (8), we get:

$$\widehat{\boldsymbol{\theta}}_K = \widetilde{\boldsymbol{\theta}}_* + \mathbf{Q}^K\mathbf{e}, \text{ with } \mathbf{Q} \text{ given by Eq. (9)} \quad (11)$$

for FLOW. The non-zero eigenvalues of $\mathbf{P}$ are $\mp\rho$ and the corresponding eigenvectors are $\frac{1}{\sqrt{2}}\big(\overline{\mathbf{e}} \pm \overline{\mathbf{e}}_\perp\big)$. Using this in (10) and simplifying, we get for vanilla FT:

$$\overline{\boldsymbol{\theta}}_K = \widetilde{\boldsymbol{\theta}}_* + \rho^K\big(\mathbb{1}(K \text{ is even})\mathbf{e} - \mathbb{1}(K \text{ is odd})\|\mathbf{e}\|_2\overline{\mathbf{e}}_\perp\big). \quad (12)$$

**Remark 7.3 (Vanilla FT).** *Since* $\rho < 1$, $\overline{\boldsymbol{\theta}}_K$ *converges to* $\widetilde{\boldsymbol{\theta}}_*$ *rapidly, and we cannot impede this convergence.*

Note that (we use $\boldsymbol{\Sigma} \succeq \mathbf{I}_d$ below):

$$\mathrm{err}_{\mathrm{tot}}(\widetilde{\boldsymbol{\theta}}_*) = \mathrm{err}_1(\widetilde{\boldsymbol{\theta}}_*) = \mathbf{e}^\top\boldsymbol{\Sigma}\mathbf{e} \geq \|\mathbf{e}\|_2^2. \quad (13)$$

On the other hand, the non-zero eigenvalues and corresponding eigenvectors of $\mathbf{Q}$ are not as straightforward to compute. We do this computation in Lemma F.3 with the re-parameterization of $\mu = \sqrt{\frac{\beta(1-\rho^2)}{(1+\beta)(1-\beta\rho^2)}}$ for some $\beta \in (0, 1]$.[5] Using this in Eq. (11) and simplifying, we get for FLOW:

$$\widehat{\boldsymbol{\theta}}_K = \widetilde{\boldsymbol{\theta}}_* + \left(\frac{\widehat{\lambda}_1^K + \widehat{\lambda}_2^K\beta^2\rho^2}{1 + \beta^2\rho^2}\right)\mathbf{e} - \beta\rho\left(\frac{\widehat{\lambda}_1^K - \widehat{\lambda}_2^K}{1 + \beta^2\rho^2}\right)\|\mathbf{e}\|\overline{\mathbf{e}}_\perp, \quad (14)$$

where $\widehat{\lambda}_1 = \frac{1+\beta\rho^2}{1+\beta}$ and $\widehat{\lambda}_2 = \rho^2\left(\frac{1-\beta}{1-\beta\rho^2}\right)$.

**Remark 7.4 (FLOW's trajectory).** *Note that we can control* $\widehat{\lambda}_1$ *by varying* $\beta$. *Specifically, we can make* $\widehat{\lambda}_1$ *arbitrarily close to* 1 *by choosing a small enough* $\beta$. *On the other hand,* $\frac{1-\beta}{1-\beta\rho^2} < \frac{1+\beta\rho^2}{1+\beta} = \widehat{\lambda}_1$ *and so,* $\widehat{\lambda}_2 < \rho^2\widehat{\lambda}_1$. *Hence, beyond a certain number of iterations* $K$, *Eq. (14) becomes:*

$$\widehat{\boldsymbol{\theta}}_K \approx \boldsymbol{\theta}_K := \widetilde{\boldsymbol{\theta}}_* + \gamma(K, \beta)\big(\mathbf{e} - \beta\rho\|\mathbf{e}\|_2\overline{\mathbf{e}}_\perp\big), \quad (15)$$

*with* $\gamma(K, \beta) := \left(\frac{\widehat{\lambda}_1^K}{1+\beta^2\rho^2}\right)$. *Because we can control* $\widehat{\lambda}_1$ *by varying* $\beta$, *we can control* $\gamma(K, \beta)$. *Thus, **we can stall convergence along*** $\big(\mathbf{e} - \beta\rho\|\mathbf{e}\|_2\overline{\mathbf{e}}_\perp\big)$,[6] ***impeding the convergence of*** $\widehat{\boldsymbol{\theta}}_K$ ***to*** $\widetilde{\boldsymbol{\theta}}_*$.

**Remark 7.5 (FLOW goes beyond model averaging).** *If we perform model averaging between* $\boldsymbol{\theta}_*$ *and* $\widetilde{\boldsymbol{\theta}}_*$ *with parameter* $\omega \in [0, 1]$, *then our averaged model is:*

$$\boldsymbol{\theta}_{\mathrm{avg}}(\omega) = \omega\boldsymbol{\theta}_* + (1 - \omega)\widetilde{\boldsymbol{\theta}}_* = \widetilde{\boldsymbol{\theta}}_* + \omega\mathbf{e}. \quad (16)$$

*Comparing the above with Eq. (15), we see that* FLOW *goes beyond model averaging because of the component along* $\overline{\mathbf{e}}_\perp$. *But we can make* $\boldsymbol{\theta}_K$ *(Eq. (15))* $\to \boldsymbol{\theta}_{\mathrm{avg}}(\omega)$ *by choosing* $\beta \to 0$ *and* $K$ *such that* $\gamma(K, \beta) \to \omega$. *So, we expect* FLOW *to be at least as powerful as model averaging.*

As per Lemma F.4, the minimum total error on both tasks with optimally tuned model averaging is given by:

$$\min_{\omega \in [0,1]} \mathrm{err}_{\mathrm{tot}}\big(\boldsymbol{\theta}_{\mathrm{avg}}(\omega)\big) = \left(\frac{\overline{\mathbf{e}}^\top\boldsymbol{\Sigma}\overline{\mathbf{e}}}{\overline{\mathbf{e}}^\top\boldsymbol{\Sigma}\overline{\mathbf{e}} + 1}\right)\|\mathbf{e}\|_2^2 < \|\mathbf{e}\|_2^2, \quad (17)$$

where recall that $\boldsymbol{\Sigma}$ is the covariance matrix of the pre-training data. On the other hand, using Eqs. (5) and (13)

$$\min\left(\mathrm{err}_{\mathrm{tot}}(\boldsymbol{\theta}_*), \mathrm{err}_{\mathrm{tot}}(\widetilde{\boldsymbol{\theta}}_*)\right) = \|\mathbf{e}\|_2^2. \quad (18)$$

---

[5] The corresponding temperature is $\tau = \frac{2\beta(1-\rho^2)\|\mathbf{e}\|_2^2}{(1-\beta^2\rho^2)}$.

[6] This direction is the top eigenvector of $\mathbf{Q}$. Since $\widetilde{\boldsymbol{\Sigma}}' = \mu(\mathbf{I}_d - \mathbf{Q})$, this is also the eigenvector of $\widetilde{\boldsymbol{\Sigma}}'$ with the smallest eigenvalue.

**Remark 7.6** (**Error comparison**). *By comparing Eqs. (17) and (18), we see that optimally tuned model averaging attains a smaller total error than both $\theta_*$ (i.e., the pre-trained model) and $\widetilde{\theta}_*$ to which vanilla FT converges rapidly (Remark 7.3). More importantly, following our discussion in Remark 7.5, we conclude that **optimally tuned FLOW's total error is at least as good as the one in Eq. (17)**.*

## 8. Conclusion

In this paper, we studied the problem of catastrophic forgetting in pre-trained models during fine-tuning when we do not have access to the pre-training data. To mitigate this issue, we proposed FLOW, a method which upweights easy samples based on the pre-trained loss values. Empirically, we showed that FLOW, on average, outperforms relevant baselines and is also complementary to these baselines in both vision and language settings. We also theoretically analyzed FLOW for linear models.

**Discussion and limitations.** We would like to conclude with an overview of our work's limitations and potential future directions. In lay terms, mitigating forgetting of pre-training capabilities comes at the cost of relatively lower fine-tuning performance. FLOW maintains this balance by sacrificing performance on *hard samples from the fine-tuning data*. Table 5 indicates that FLOW has lower accuracy on samples with high pre-training loss ("hard samples") compared to standard FT. Our method selectively downweighs samples with high pre-training losses for preserving pre-training performance. An interesting future direction is improving performance on such samples while maintaining or improving overall performance. On the theoretical side, we hope to extend our analysis to generalized linear models (GLMs) and even non-linear models.

*Table 5.* **Comparison of FLOW and Standard-FT on hard samples across three vision datasets.** We evaluate performance on the top 10% hardest samples (those with the highest pre-trained losses). Indeed, the samples with high pre-training losses have lower accuracy when using FLOW compared to standard fine-tuning (FT). This is an unsurprising outcome of our approach; we sacrifice performance on hard examples of the fine-tuning data to maintain performance on the pre-training data.

| Dataset | # of Hard Samples | Standard FT | FLOW |
|---|---|---|---|
| CIFAR-10 | 1000 | 86.60 | 30.70 |
| CIFAR-100 | 1000 | 56.40 | 21.30 |
| Stanford Cars | 805 | 71.30 | 13.18 |

## Acknowledgments

This research was supported by NSF EnCORE Tripods (2217069) and NSF AI Institute for the Foundations of Machine Learning (2019844). Research of Ali Kavis is funded in part by the Swiss National Science Foundation (SNSF) under grant number P500PT_217942. The authors are grateful to anonymous reviewers for their feedback on improving this paper.

## Impact Statement

This paper presents work whose goal is to advance the field of machine learning. There are potential societal consequences of our work, none of which we feel must be specifically highlighted here.

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

# Appendix

## Table of Contents

## A. Extended Related Work

The majority of the approaches for mitigating forgetting assume task-specific knowledge access to different extents; either (a subset of) the pre-training dataset itself or some information/statistic computed from pre-training data. Below, we describe the data-aware approaches based on how they make use of task-specific knowledge.

**Regularization-based methods.** This line of work aims to preserve performance on previously learned tasks by keeping the (fine-tuned) model parameters close to the pre-trained model. The key idea is to introduce task-specific regularization in the fine-tuning phase which will penalize updates along the "important" directions for the old tasks (Ahn et al., 2019). Kirkpatrick et al. (2016) introduces the elastic weight consolidation (EWC) algorithm, which estimates the important direction per-task by calculating a diagonal approximation to the Fisher information matrix (FIM), which acts as the weight matrix for the regularization term. Several variants of EWC have been subsequently proposed (Schwarz et al., 2018; Ritter et al., 2018; Lee et al., 2020; Liu et al., 2018). Zenke et al. (2017); Aljundi et al. (2018) adopt online strategies to infer the importance of each parameter by their variational effect on the model outputs. In a spirit similar to EWC, Lee et al. (2017) incrementally matches the posterior of the pre-trained model and the new task by assuming Gaussian posteriors.

**Optimization-driven methods.** Another perspective to mitigating forgetting is guiding the optimization process by constraining the algorithms directly as opposed to manipulating the loss function. The core idea is to keep track of "important directions" for the old tasks, and train on the new task "orthogonally." This could be done by storing prior data samples or gradients in a buffer (Lopez-Paz & Ranzato, 2017; Farajtabar et al., 2020; Chaudhry et al., 2019a) or by incrementally expanding the subspace of important directions without storing task-specific information (Zeng et al., 2019; Wang et al., 2021; 2023b).

**Replay-based methods.** Drawing inspiration from the complementary learning systems theory (Mcclelland et al., 1995), a more direct approach is to introduce samples from old tasks into the training process for the new task. Samples are selected in a streaming fashion or by manually crafting a subset on demand, stored in dedicated buffers and replayed during the fine-tuning. The intuition is that the task-specific representations are refreshed periodically through historical data.

Replay-based methods consist of two fundamental components: data selection and data reiteration mechanisms. When the data is received in a streaming fashion, information has to be buffered online (Riemer et al., 2019; Chaudhry et al., 2019b; Isele & Cosgun, 2018; De Lange & Tuytelaars, 2021). In the case when datasets are available on demand, Rebuffi et al. (2017) selects samples which are "representative" of their respective class, while others focus on inducing diversity (Aljundi et al., 2019; Bang et al., 2021) and balance (Borsos et al., 2020; Tiwari et al., 2021) across buffered data. For the scenarios in which storage is limited, Caccia et al. (2020); Wang et al. (2022b) develop compression methods for buffered data. As a complimentary component to the data selection process, how the buffered data is replayed plays a significant role in the success of such methods. A fundamental idea, which has several interpretations across the board, is knowledge distillation (Hinton et al., 2015). Prior work argues that augmenting fine-tuning with knowledge distillation shows great performance on the forgetting front (Lopez-Paz & Ranzato, 2017; Chaudhry et al., 2019a; Rebuffi et al., 2017; Jung et al., 2017; Triki et al., 2017; Li & Hoiem, 2016; Lee et al., 2019b; Dhar et al., 2019).

An orthogonal research direction focuses on maintaining a generative model that could reliably output pseudo-samples that are representative of the dataset of the old tasks (Kemker & Kanan, 2018; Wu et al., 2018). Note that generative approaches are prone to scalability issues and distribution shifts.

**Architecture-driven methods.** Another technique to limit the interference between tasks is allocating a separate trainable set of parameters per task. This could be done by initializing a sub-networks per new task (Rusu et al., 2016; Aljundi et al., 2017; Collier et al., 2020; Rajasegaran et al., 2019; Ramesh & Chaudhari, 2021; Wang et al., 2023a; 2022a), gradually expanding the parameters of a base network (Yoon et al., 2018; Ostapenko et al., 2019; Hung et al., 2019), or segregating a fixed model into task-specific subset of parameters (Mallya et al., 2018; Kang et al., 2022; Serra et al., 2018; Wortsman et al., 2020; Mallya & Lazebnik, 2017; Mustafa B Gurbuz, 2022; Jung et al., 2020). While some parameters are task-specific, parts of the overall model could be shared to enable knowledge transfer. The main downside associated is that the task identity must be available during inference to (de)activate relevant sub-networks, hindering versatility. Aljundi et al. (2017) develop dedicated strategies to overcome the need for task identification by automatizing task-specific parameter activation.

**FLOW and its connection to other ML applications.** Our approach has connections to factuality in LLM training (Ghosal et al., 2024; Gekhman et al., 2024). For a fine-tuning task on a factual knowledge-based dataset, samples that are not properly represented in the pre-training distribution could be considered hard. In the data-oblivious setting, one principled way would be to rank *hardness* based on pre-training loss values; therefore, FLOW would identify such samples and guide

the fine-tuning process such that those samples will not unexpectedly deteriorate the performance.

Although fundamentally different in its objective, machine unlearning is another avenue of application for our sample weighting approach which could be interpreted as a means of *soft unlearning*. While unlearning and forgetting sound similar, they are not the same in spirit; in unlearning we deliberately induce "forgetting" on some samples, whereas in our context, forgetting is an undesirable side effect that we want to avoid. Hence, it might not be straightforward extend techniques of one into another, one could extend our sample-wise weighting scheme with appropriate modifications to help selectively unlearn specific samples.

## B. Our Algorithm in the Presence of Task-Specific Model Components

Suppose our model is parameterized by $\boldsymbol{\theta} = \mathbf{U} \cup \mathbf{V}$, where $\mathbf{U}$ is the common/shared part of the model for all tasks (i.e., this part remains the same for all tasks), and $\mathbf{V}$ is the task-specific part of the model. In particular, in our vision experiments, the models have task-specific prediction heads (i.e., softmax layers) and batch-norm (BN). The modified version of Algorithm 1 in the presence of task-specific components is stated in Algorithm 2. The main differences from Algorithm 1 are steps (i) and (iv) – these steps optimize the task-specific part for the new task with uniform weighting. It is worth mentioning that if our model consists of task-specific prediction heads – which is the case in our vision experiments – then steps (i) and (iv) are just vanilla linear probing with the pre-trained body and the body learned after fine-tuning, respectively.

---

**Algorithm 2** Fine-tuning with Pre-trained **L**oss-**O**riented **W**eighting (**FLOW**)

---

**Input:** Pre-trained model $\boldsymbol{\theta}_*^{(1)} = \mathbf{U}_*^{(1)} \cup \mathbf{V}_*^{(1)}$, dataset $\{(\mathbf{x}_i, \mathbf{y}_i)\}_{i=1}^n$ for the new task, and temperature parameter $\tau$.

$f_i(\mathbf{U}, \mathbf{V}) \to i^{\text{th}}$ sample's loss at $\boldsymbol{\theta} = \mathbf{U} \cup \mathbf{V}$, with a non-negative loss function (e.g., cross-entropy loss).

**Step (i)** Fine-tune task-specific part for new task with vanilla unweighted loss: $\mathbf{V}_*^{(2)} := \text{argmin}_{\mathbf{V}} \sum_{i=1}^n f_i(\mathbf{U}_*^{(1)}, \mathbf{V})$.

**Step (ii)** Compute sample weights: $w_i = \exp\left(-f_i(\mathbf{U}_*^{(1)}, \mathbf{V}_*^{(2)})/\tau\right)$.

**Step (iii)** Fine-tune full model **with weighted loss**: $\overline{\mathbf{U}}_*^{(2)}, \overline{\mathbf{V}}_*^{(2)} := \text{argmin}_{\mathbf{U}, \mathbf{V}} \sum_{i=1}^n w_i f_i(\mathbf{U}, \mathbf{V})$.

**Step (iv)** Fine-tune task-specific part for new task *using the learned common part* with vanilla unweighted loss: $\widehat{\mathbf{V}}_*^{(2)} := \text{argmin}_{\mathbf{V}} \sum_{i=1}^n f_i(\overline{\mathbf{U}}_*^{(2)}, \mathbf{V})$.

**Output:** New model for

- Original/pre-training task is $\widehat{\boldsymbol{\theta}}_*^{(1)} = \overline{\mathbf{U}}_*^{(2)} \cup \mathbf{V}_*^{(1)}$.

- New/fine-tuning task is $\widehat{\boldsymbol{\theta}}_*^{(2)} = \overline{\mathbf{U}}_*^{(2)} \cup \widehat{\mathbf{V}}_*^{(2)}$.

---

**Remark B.1.** *In all our vision experiments (with task-specific parts), we set $\tau = \text{median}(f_i(\mathbf{U}_*^{(1)}, \mathbf{V}_*^{(2)}))$ (similar to Remark 4.2).*

## C. Proof of Proposition 4.3

*Proof.* We wish to minimize $g(\boldsymbol{\pi}) = \sum_{i=1}^n \pi_i f_i(\boldsymbol{\theta}^*) + \tau \sum_{i=1}^n \pi_i \log \pi_i$ subject to $\sum_{i=1}^n \pi_i = 1$ and $\pi_i \geq 0$ for all $i \in [n]$. The proof is a straightforward application of Lagrangian multipliers. It is enough to enforce $\sum_{i=1}^n \pi_i = 1$ only ($\pi_i \geq 0$ for all $i \in [n]$ will also follow). For that, the Lagrangian function is:

$$J(\boldsymbol{\pi}, \lambda) = \sum_{i=1}^n \pi_i f_i(\boldsymbol{\theta}^*) + \tau \sum_{i=1}^n \pi_i \log \pi_i + \lambda\left(\sum_{i=1}^n \pi_i - 1\right), \tag{19}$$

where $\lambda$ is the Lagrangian multiplier. Now, at the optimal point $\boldsymbol{\pi}^* = [\pi_1^*, \ldots, \pi_n^*]^\top$, we must have:

$$\frac{\partial J}{\partial \pi_i}\bigg|_{\pi_i^*} = f_i(\boldsymbol{\theta}^*) + \tau\left(1 + \log \pi_i^*\right) + \lambda = 0, \tag{20}$$

for all $i \in [n]$. Simplifying, we get:

$$\pi_i^* = \frac{1}{Z} \exp\left(-\frac{f_i(\boldsymbol{\theta}^*)}{\tau}\right), \tag{21}$$

where $Z = \exp\left(\left(1 + \frac{\lambda}{\tau}\right)\right)$ is the normalizing constant. To have $\sum_{i=1}^n \pi_i = 1$, we get $Z = \sum_{j=1}^n \exp\left(-\frac{f_j(\mathbf{W}^*)}{\tau}\right)$. Also, note that we are good with the non-negativity constraints. $\qquad\square$

## D. Proof of Theorem 7.2

*Proof.* Note that:

$$\widehat{\mathrm{err}}_2(\widehat{\boldsymbol{\theta}}) = \left(\widehat{\boldsymbol{\theta}} - \widetilde{\boldsymbol{\theta}}_*\right)^\top \mathbb{E}_{\widetilde{\mathcal{D}}}\left[w(\widetilde{\mathbf{x}}, \widetilde{\mathbf{y}})\widetilde{\mathbf{x}}\widetilde{\mathbf{x}}^\top\right]\left(\widehat{\boldsymbol{\theta}} - \widetilde{\boldsymbol{\theta}}_*\right). \tag{22}$$

Also, after plugging in $\widetilde{\mathbf{y}} = \langle \widetilde{\boldsymbol{\theta}}_*, \widetilde{\mathbf{x}}\rangle$, we get:

$$w(\widetilde{\mathbf{x}}, \widetilde{\mathbf{y}}) = \exp\left(-\frac{\left(\langle \boldsymbol{\theta}_* - \widetilde{\boldsymbol{\theta}}_*, \widetilde{\mathbf{x}}\rangle\right)^2}{\tau}\right).$$

Recall $\mathbf{e} := \boldsymbol{\theta}_* - \widetilde{\boldsymbol{\theta}}_*$ and $\overline{\mathbf{e}} := \frac{\mathbf{e}}{\|\mathbf{e}\|_2}$. Suppose $\tau = \alpha\|\mathbf{e}\|_2^2$, for some $\alpha > 0$. Then $w(\widetilde{\mathbf{x}}, \widetilde{\mathbf{y}}) = \exp\left(-\frac{(\langle \overline{\mathbf{e}}, \widetilde{\mathbf{x}}\rangle)^2}{\alpha}\right)$, and we can focus on

$$\widetilde{\boldsymbol{\Sigma}}' := \mathbb{E}_{\widetilde{\mathbf{x}}\sim\widetilde{\mathcal{P}}}\left[\exp\left(-\frac{\left(\langle \overline{\mathbf{e}}, \widetilde{\mathbf{x}}\rangle\right)^2}{\alpha}\right)\widetilde{\mathbf{x}}\widetilde{\mathbf{x}}^\top\right]. \tag{23}$$

Let $\mu = \left(\frac{\alpha}{\alpha+2}\right)^{1/2} = \left(\frac{\tau}{\tau+2\|\mathbf{e}\|_2^2}\right)^{1/2}$. As per Lemma F.1, we have:

$$\widetilde{\boldsymbol{\Sigma}}' = \mu\left(\mathbf{I}_d - \mathbf{Q}\right), \tag{24}$$

where

$$\mathbf{Q} = (1 - \mu^2)\overline{\mathbf{e}}\,\overline{\mathbf{e}}^\top + \rho^2(1 - \mu^2)\overline{\mathbf{e}}_\perp\overline{\mathbf{e}}_\perp^\top - \rho\mu^2\left(\overline{\mathbf{e}}\,\overline{\mathbf{e}}_\perp^\top + \overline{\mathbf{e}}_\perp\overline{\mathbf{e}}^\top\right). \tag{25}$$

So if we minimize $\widehat{\mathrm{err}}_2(\widehat{\boldsymbol{\theta}})$ with GD starting from $\widehat{\boldsymbol{\theta}}_0 = \boldsymbol{\theta}_*$ and using a constant learning rate $\widehat{\eta}$, our iterate $\widehat{\boldsymbol{\theta}}_K$ at the $K^{\text{th}}$ iteration satisfies:

$$\widehat{\boldsymbol{\theta}}_K - \widetilde{\boldsymbol{\theta}}_* = \left(\mathbf{I}_d - 2\widehat{\eta}\widetilde{\boldsymbol{\Sigma}}'\right)^K\left(\boldsymbol{\theta}_* - \widetilde{\boldsymbol{\theta}}_*\right) = \left(\mathbf{I}_d - 2\widehat{\eta}\widetilde{\boldsymbol{\Sigma}}'\right)^K\mathbf{e}, \tag{26}$$

where the last step follows by recalling that $\boldsymbol{\theta}_* - \widetilde{\boldsymbol{\theta}}_* = \mathbf{e}$, and $\widetilde{\boldsymbol{\Sigma}}'$ is given by Equation (24). $\qquad\square$

## E. Difficulty in the Analysis with a General Covariance Matrix $\widetilde{\boldsymbol{\Sigma}}$

We will first derive the *weighted* (fine-tuning) data covariance matrix $\widetilde{\boldsymbol{\Sigma}}'$ in the context of Theorem 7.2 for a general (fine-tuning) data covariance matrix $\widetilde{\boldsymbol{\Sigma}}$. Specifically, following the proof of Theorem 7.2, we have:

$$\widetilde{\boldsymbol{\Sigma}}' := \mathbb{E}_{\widetilde{\mathbf{x}}\sim\mathcal{N}(\vec{\mathbf{0}}_d, \widetilde{\boldsymbol{\Sigma}})}\left[\exp\left(-\frac{\left(\langle \mathbf{e}, \widetilde{\mathbf{x}}\rangle\right)^2}{\tau}\right)\widetilde{\mathbf{x}}\widetilde{\mathbf{x}}^\top\right]. \tag{27}$$

Note that $\widetilde{\mathbf{x}} = \widetilde{\boldsymbol{\Sigma}}^{1/2}\mathbf{z}$, where $\mathbf{z} \sim \mathcal{N}(\vec{\mathbf{0}}_d, \mathbf{I}_d)$. Using this above, we get:

$$\widetilde{\boldsymbol{\Sigma}}' = \widetilde{\boldsymbol{\Sigma}}^{1/2}\mathbb{E}\left[\exp\left(-\frac{\left(\langle \mathbf{e}, \widetilde{\boldsymbol{\Sigma}}^{1/2}\mathbf{z}\rangle\right)^2}{\tau}\right)\mathbf{z}\mathbf{z}^\top\right]\widetilde{\boldsymbol{\Sigma}}^{1/2} = \widetilde{\boldsymbol{\Sigma}}^{1/2}\mathbb{E}\left[\exp\left(-\frac{\left(\langle \widetilde{\boldsymbol{\Sigma}}^{1/2}\mathbf{e}, \mathbf{z}\rangle\right)^2}{\tau}\right)\mathbf{z}\mathbf{z}^\top\right]\widetilde{\boldsymbol{\Sigma}}^{1/2}, \tag{28}$$

where the last step follows by using the symmetry of $\widetilde{\boldsymbol{\Sigma}}$. Let $\tau = \alpha\left\|\widetilde{\boldsymbol{\Sigma}}^{1/2}\mathbf{e}\right\|_2^2$ for some $\alpha > 0$. Also, let $\mathbf{r} := (\widetilde{\boldsymbol{\Sigma}}^{1/2}\mathbf{e})/\|\widetilde{\boldsymbol{\Sigma}}^{1/2}\mathbf{e}\|_2$. In that case, we have:

$$\widetilde{\boldsymbol{\Sigma}}' = \widetilde{\boldsymbol{\Sigma}}^{1/2}\mathbf{M}\widetilde{\boldsymbol{\Sigma}}^{1/2}, \text{ where } \mathbf{M} := \mathbb{E}\left[\exp\left(-\frac{\left(\langle \mathbf{r}, \mathbf{z}\rangle\right)^2}{\alpha}\right)\mathbf{z}\mathbf{z}^\top\right]. \tag{29}$$

Suppose $\{\mathbf{r}_{\perp,j}\}_{j=1}^{d-1}$ is an orthonormal basis for the subspace of $\mathbb{R}^d$ orthogonal to $\mathbf{r}$; so $\langle \mathbf{r}_{\perp,j}, \mathbf{r} \rangle = 0 \; \forall \, j \in [d-1]$ and $\langle \mathbf{r}_{\perp,j}, \mathbf{r}_{\perp,k} \rangle = \mathbb{1}(j = k) \; \forall \, j, k \in [d-1]$. Note that $\{\mathbf{r}, \mathbf{r}_{\perp,1}, \ldots, \mathbf{r}_{\perp,d-1}\}$ forms an orthonormal basis for $\mathbb{R}^d$. Then, as per Lemma F.5, we have that $\mathbf{r}$ is an eigenvector of $\mathbf{M}$ with eigenvalue $\left(\frac{\alpha}{\alpha+2}\right)^{3/2}$, and each $\mathbf{r}_{\perp,j}$ is an eigenvector of $\mathbf{M}$ with eigenvalue $\left(\frac{\alpha}{\alpha+2}\right)^{1/2}$. For brevity, let $\mu = \left(\frac{\alpha}{\alpha+2}\right)^{1/2}$. Then, we can write:

$$\mathbf{M} = \mu^3 \mathbf{r}\mathbf{r}^\top + \mu \sum_{j=1}^{d-1} \mathbf{r}_{\perp,j}\mathbf{r}_{\perp,j}^\top = \mu^3 \mathbf{r}\mathbf{r}^\top + \mu\left(\mathbf{I}_d - \mathbf{r}\mathbf{r}^\top\right), \tag{30}$$

where the last step follows because $\{\mathbf{r}, \mathbf{r}_{\perp,1}, \ldots, \mathbf{r}_{\perp,d-1}\}$ forms an orthonormal basis for $\mathbb{R}^d$, due to which $\mathbf{r}\mathbf{r}^\top + \sum_{j=1}^{d-1} \mathbf{r}_{\perp,j}\mathbf{r}_{\perp,j}^\top = \mathbf{I}_d$. Simplifying Equation (30) a bit, we get:

$$\mathbf{M} = \mu\left(\mathbf{I}_d - (1 - \mu^2)\mathbf{r}\mathbf{r}^\top\right). \tag{31}$$

Plugging this into Equation (29) and recalling that $\mathbf{r} := (\widetilde{\boldsymbol{\Sigma}}^{1/2}\mathbf{e})/\|\widetilde{\boldsymbol{\Sigma}}^{1/2}\mathbf{e}\|_2$, we get:

$$\widetilde{\boldsymbol{\Sigma}}' = \mu\mathbf{B}, \text{ where } \mathbf{B} := \left(\widetilde{\boldsymbol{\Sigma}} - (1-\mu^2)\frac{\widetilde{\boldsymbol{\Sigma}}\mathbf{e}\mathbf{e}^\top\widetilde{\boldsymbol{\Sigma}}}{\mathbf{e}^\top\widetilde{\boldsymbol{\Sigma}}\mathbf{e}}\right). \tag{32}$$

Equation (32) is the weighted covariance matrix for a general $\widetilde{\boldsymbol{\Sigma}}$.

**Remark E.1** (**Difficulty with general $\widetilde{\boldsymbol{\Sigma}}$**). *It is hard to proceed with the analysis after this point because it is difficult to characterize the eigen-spectrum of $\mathbf{B}$ in general, without assuming any relation between $\widetilde{\boldsymbol{\Sigma}}$ and $\mathbf{e}$. This is what we meant in Remark 7.1.*

# F. Lemmas Used and Their Proofs

**Lemma F.1.** *In the proof of Theorem 7.2, recall that $\tau = \alpha\|\mathbf{e}\|_2^2$. Then, we have:*

$$\widetilde{\boldsymbol{\Sigma}}' := \mathbb{E}_{\widetilde{\mathbf{x}} \sim \widetilde{\mathcal{P}}}\left[\exp\left(-\frac{(\langle \overline{\mathbf{e}}, \widetilde{\mathbf{x}}\rangle)^2}{\alpha}\right)\widetilde{\mathbf{x}}\widetilde{\mathbf{x}}^\top\right] = \mu\left(\mathbf{I}_d - (1-\mu^2)\overline{\mathbf{e}}\,\overline{\mathbf{e}}^\top - \rho^2(1-\mu^2)\overline{\mathbf{e}}_\perp\overline{\mathbf{e}}_\perp^\top + \rho\mu^2\left(\overline{\mathbf{e}}\,\overline{\mathbf{e}}_\perp^\top + \overline{\mathbf{e}}_\perp\overline{\mathbf{e}}^\top\right)\right),$$

*where $\mu = \left(\frac{\alpha}{\alpha+2}\right)^{1/2} = \left(\frac{\tau}{\tau+2\|\mathbf{e}\|_2^2}\right)^{1/2}$.*

*Proof.* Recall that $\overline{\mathbf{e}}$ and $\overline{\mathbf{e}}_\perp$ are orthogonal to each other and both are unit-norm. Suppose $\{\overline{\mathbf{e}}_{\perp,3}, \overline{\mathbf{e}}_{\perp,4}, \ldots, \overline{\mathbf{e}}_{\perp,d}\}$ is an orthonormal basis for the $(d-2)$-dimensional subspace of $\mathbb{R}^d$ orthogonal to $\overline{\mathbf{e}}$ and $\overline{\mathbf{e}}_\perp$. Thus, $\{\overline{\mathbf{e}}, \overline{\mathbf{e}}_\perp, \overline{\mathbf{e}}_{\perp,3}, \overline{\mathbf{e}}_{\perp,4}, \ldots, \overline{\mathbf{e}}_{\perp,d}\}$ is an orthonormal basis for $\mathbb{R}^d$. Then using Lemma F.2, we can write:

$$\widetilde{\mathbf{x}} = \mathrm{z}_1\overline{\mathbf{e}} + \left(\rho\mathrm{z}_1 + \sqrt{1-\rho^2}\mathrm{z}_2\right)\overline{\mathbf{e}}_\perp + \sum_{j=3}^{d} \mathrm{z}_j\overline{\mathbf{e}}_{\perp,j}, \tag{33}$$

where $\{\mathrm{z}_j\}_{j=1}^{d} \underset{\mathrm{iid}}{\sim} \mathcal{N}(0,1)$.

Using independence and zero-mean nature of $\{\mathrm{z}_j\}_{j=1}^{d}$, we get:

$$\widetilde{\boldsymbol{\Sigma}}' = \underbrace{\mathbb{E}\left[\exp\left(-\frac{\mathrm{z}_1^2}{\alpha}\right)\mathrm{z}_1^2\right]}_{:=T_1}\overline{\mathbf{e}}\,\overline{\mathbf{e}}^\top + \underbrace{\mathbb{E}\left[\exp\left(-\frac{\mathrm{z}_1^2}{\alpha}\right)\mathrm{z}_1\left(\rho\mathrm{z}_1 + \sqrt{1-\rho^2}\mathrm{z}_2\right)\right]}_{:=T_2}\left(\overline{\mathbf{e}}\,\overline{\mathbf{e}}_\perp^\top + \overline{\mathbf{e}}_\perp\overline{\mathbf{e}}^\top\right)$$

$$+ \underbrace{\mathbb{E}\left[\exp\left(-\frac{\mathrm{z}_1^2}{\alpha}\right)\left(\rho\mathrm{z}_1 + \sqrt{1-\rho^2}\mathrm{z}_2\right)^2\right]}_{:=T_3}\overline{\mathbf{e}}_\perp\overline{\mathbf{e}}_\perp^\top + \sum_{j=3}^{d} \underbrace{\mathbb{E}\left[\exp\left(-\frac{\mathrm{z}_1^2}{\alpha}\right)\right]}_{:=T_4}\underbrace{\mathbb{E}\left[\mathrm{z}_j^2\right]}_{=1}\overline{\mathbf{e}}_{\perp,j}\overline{\mathbf{e}}_{\perp,j}^\top. \tag{34}$$

Note that (we use the independence of $z_1$ and $z_2$):

$$T_2 = \rho T_1 + \sqrt{1-\rho^2}\mathbb{E}\Big[\exp\Big(-\frac{z_1^2}{\alpha}\Big)z_1\Big]\underbrace{\mathbb{E}[z_2]}_{=0} = \rho T_1, \tag{35}$$

and

$$T_3 = \rho^2 T_1 + 2\rho\sqrt{1-\rho^2}\Big[\exp\Big(-\frac{z_1^2}{\alpha}\Big)z_1\Big]\underbrace{\mathbb{E}[z_2]}_{=0} + (1-\rho^2)T_4\underbrace{\mathbb{E}[z_2^2]}_{=1} = \rho^2 T_1 + (1-\rho^2)T_4. \tag{36}$$

In the above two equations, we have again used the independence of $z_1$ and $z_2$. Now we will compute $T_1$ and $T_4$. We have:

$$T_1 = \left(\frac{1}{\sqrt{2\pi}}\int_{-\infty}^{\infty} z_1^2\exp\Big(-z_1^2\Big(\frac{1}{\alpha}+\frac{1}{2}\Big)\Big)dz_1\right) = \Big(\frac{\alpha}{\alpha+2}\Big)^{3/2}, \tag{37}$$

and

$$T_4 = \left(\frac{1}{\sqrt{2\pi}}\int_{-\infty}^{\infty}\exp\Big(-z_1^2\Big(\frac{1}{\alpha}+\frac{1}{2}\Big)\Big)dz_1\right) = \Big(\frac{\alpha}{\alpha+2}\Big)^{1/2}. \tag{38}$$

Recall that $\mu = \Big(\frac{\alpha}{\alpha+2}\Big)^{1/2}$. Plugging this into Equations (35) to (38) gives us:

$$T_1 = \mu^3, T_2 = \rho\mu^3, T_3 = \rho^2\mu^3 + (1-\rho^2)\mu, \text{ and } T_4 = \mu. \tag{39}$$

Plugging this into Equation (34) gives us:

$$\widetilde{\boldsymbol{\Sigma}}' = \mu^3\overline{\mathbf{e}}\overline{\mathbf{e}}^\top + \rho\mu^3\big(\overline{\mathbf{e}}\overline{\mathbf{e}}_\perp^\top + \overline{\mathbf{e}}_\perp\overline{\mathbf{e}}^\top\big) + \Big(\rho^2\mu^3 + (1-\rho^2)\mu\Big)\overline{\mathbf{e}}_\perp\overline{\mathbf{e}}_\perp^\top + \mu\sum_{j=3}^{d}\overline{\mathbf{e}}_{\perp,j}\overline{\mathbf{e}}_{\perp,j}^\top. \tag{40}$$

Recall that $\{\overline{\mathbf{e}}, \overline{\mathbf{e}}_\perp, \overline{\mathbf{e}}_{\perp,3}, \overline{\mathbf{e}}_{\perp,4}, \ldots, \overline{\mathbf{e}}_{\perp,d}\}$ is an orthonormal basis for $\mathbb{R}^d$. Thus, $\sum_{j=3}^{d}\overline{\mathbf{e}}_{\perp,j}\overline{\mathbf{e}}_{\perp,j}^\top = \mathbf{I}_d - \overline{\mathbf{e}}\overline{\mathbf{e}}^\top - \overline{\mathbf{e}}_\perp\overline{\mathbf{e}}_\perp^\top$. Using this above, we get:

$$\widetilde{\boldsymbol{\Sigma}}' = \mu\Big(\mathbf{I}_d - (1-\mu^2)\overline{\mathbf{e}}\overline{\mathbf{e}}^\top - \rho^2(1-\mu^2)\overline{\mathbf{e}}_\perp\overline{\mathbf{e}}_\perp^\top + \rho\mu^2\big(\overline{\mathbf{e}}\overline{\mathbf{e}}_\perp^\top + \overline{\mathbf{e}}_\perp\overline{\mathbf{e}}^\top\big)\Big). \tag{41}$$

This finishes the proof. □

**Lemma F.2.** *Suppose* $\{\overline{\mathbf{e}}, \overline{\mathbf{e}}_\perp, \overline{\mathbf{e}}_{\perp,3}, \overline{\mathbf{e}}_{\perp,4}, \ldots, \overline{\mathbf{e}}_{\perp,d}\}$ *is an orthonormal basis for* $\mathbb{R}^d$. *If* $\widetilde{\mathbf{x}} \sim \mathcal{N}(\vec{\mathbf{0}}_d, \widetilde{\boldsymbol{\Sigma}})$, *then we can write:*

$$\widetilde{\mathbf{x}} = z_1\overline{\mathbf{e}} + \Big(\rho z_1 + \sqrt{1-\rho^2}z_2\Big)\overline{\mathbf{e}}_\perp + \sum_{j=3}^{d}z_j\overline{\mathbf{e}}_{\perp,j}, \tag{42}$$

*where* $\{z_j\}_{j=1}^{d}\underset{\text{iid}}{\sim}\mathcal{N}(0,1)$.

*Proof.* If $\widetilde{\mathbf{x}}$ is as per Equation (42), then clearly $\widetilde{\mathbf{x}}$ is a zero-mean Gaussian. All that remains to show is that

$$\mathbb{E}\Big[\widetilde{\mathbf{x}}\widetilde{\mathbf{x}}^\top\Big] = \widetilde{\boldsymbol{\Sigma}} = \mathbf{I}_d + \rho\big(\overline{\mathbf{e}}\overline{\mathbf{e}}_\perp^\top + \overline{\mathbf{e}}_\perp\overline{\mathbf{e}}^\top\big).$$

Using independence and zero-mean nature of $\{z_j\}_{j=1}^{d}$, we get:

$$\mathbb{E}\Big[\widetilde{\mathbf{x}}\widetilde{\mathbf{x}}^\top\Big] = \underbrace{\mathbb{E}[z_1^2]}_{=1}\overline{\mathbf{e}}\overline{\mathbf{e}}^\top + \underbrace{\mathbb{E}\Big[z_1\Big(\rho z_1 + \sqrt{1-\rho^2}z_2\Big)\Big]}_{:=(A)}\big(\overline{\mathbf{e}}\overline{\mathbf{e}}_\perp^\top + \overline{\mathbf{e}}_\perp\overline{\mathbf{e}}^\top\big) + \underbrace{\mathbb{E}\Big[\Big(\rho z_1 + \sqrt{1-\rho^2}z_2\Big)^2\Big]}_{:=(B)}\overline{\mathbf{e}}_\perp\overline{\mathbf{e}}_\perp^\top$$

$$+ \sum_{j=3}^{d}\underbrace{\mathbb{E}[z_j^2]}_{=1}\overline{\mathbf{e}}_{\perp,j}\overline{\mathbf{e}}_{\perp,j}^\top. \tag{43}$$

Note that (we use the independence of $z_1$ and $z_2$):

$$(\mathrm{A}) = \rho \underbrace{\mathbb{E}[z_1^2]}_{=1} + \sqrt{1 - \rho^2}\, \underbrace{\mathbb{E}[z_1]}_{=0}\, \underbrace{\mathbb{E}[z_2]}_{=0} = \rho, \tag{44}$$

and

$$(\mathrm{B}) = \rho^2 \underbrace{\mathbb{E}[z_1^2]}_{=1} + 2\rho\sqrt{1 - \rho^2}\, \underbrace{\mathbb{E}[z_1]}_{=0}\, \underbrace{\mathbb{E}[z_2]}_{=0} + (1 - \rho^2)\underbrace{\mathbb{E}[z_2^2]}_{=1} = 1. \tag{45}$$

Plugging this into Equation (43), we get:

$$\mathbb{E}\left[\widetilde{\mathbf{x}}\widetilde{\mathbf{x}}^\top\right] = \overline{\mathbf{e}}\,\overline{\mathbf{e}}^\top + \rho\left(\overline{\mathbf{e}}\,\overline{\mathbf{e}}_\perp^\top + \overline{\mathbf{e}}_\perp\overline{\mathbf{e}}^\top\right) + \overline{\mathbf{e}}_\perp\overline{\mathbf{e}}_\perp^\top + \sum_{j=3}^{d} \overline{\mathbf{e}}_{\perp,j}\overline{\mathbf{e}}_{\perp,j}^\top. \tag{46}$$

Recall that $\{\overline{\mathbf{e}}, \overline{\mathbf{e}}_\perp, \overline{\mathbf{e}}_{\perp,3}, \overline{\mathbf{e}}_{\perp,4}, \dots, \overline{\mathbf{e}}_{\perp,d}\}$ is an orthonormal basis for $\mathbb{R}^d$. Thus, $\sum_{j=3}^{d} \overline{\mathbf{e}}_{\perp,j}\overline{\mathbf{e}}_{\perp,j}^\top = \mathbf{I}_d - \overline{\mathbf{e}}\,\overline{\mathbf{e}}^\top - \overline{\mathbf{e}}_\perp\overline{\mathbf{e}}_\perp^\top$. Using this above, we get:

$$\mathbb{E}\left[\widetilde{\mathbf{x}}\widetilde{\mathbf{x}}^\top\right] = \mathbf{I}_d + \rho\left(\overline{\mathbf{e}}\,\overline{\mathbf{e}}_\perp^\top + \overline{\mathbf{e}}_\perp\overline{\mathbf{e}}^\top\right) = \widetilde{\mathbf{\Sigma}}. \tag{47}$$

This finishes the proof. $\qquad\square$

**Lemma F.3.** *Recall that*

$$\mathbf{Q} = (1 - \mu^2)\overline{\mathbf{e}}\,\overline{\mathbf{e}}^\top + \rho^2(1 - \mu^2)\overline{\mathbf{e}}_\perp\overline{\mathbf{e}}_\perp^\top - \rho\mu^2\left(\overline{\mathbf{e}}\,\overline{\mathbf{e}}_\perp^\top + \overline{\mathbf{e}}_\perp\overline{\mathbf{e}}^\top\right).$$

*Let*

$$\mu = \sqrt{\frac{\beta(1 - \rho^2)}{(1 + \beta)(1 - \beta\rho^2)}}$$

*for some $\beta \in (0, 1]$. In that case, the eigenvalues of $\mathbf{Q}$ are:*

$$\widehat{\lambda}_1 = \frac{1 + \beta\rho^2}{1 + \beta} \text{ and } \widehat{\lambda}_2 = \rho^2\left(\frac{1 - \beta}{1 - \beta\rho^2}\right),$$

*and the corresponding eigenvectors are:*

$$\widehat{\mathbf{v}}_1 = \frac{1}{\sqrt{1 + \beta^2\rho^2}}\overline{\mathbf{e}} - \frac{\beta\rho}{\sqrt{1 + \beta^2\rho^2}}\overline{\mathbf{e}}_\perp \text{ and } \widehat{\mathbf{v}}_2 = -\frac{\beta\rho}{\sqrt{1 + \beta^2\rho^2}}\overline{\mathbf{e}} - \frac{1}{\sqrt{1 + \beta^2\rho^2}}\overline{\mathbf{e}}_\perp.$$

*Proof.* $\mathbf{Q}$ is a rank-2 matrix and its two eigenvectors will be in the span of $\overline{\mathbf{e}}$ and $\overline{\mathbf{e}}_\perp$. In particular, an eigenvector of $\mathbf{Q}$ is of the form $[\overline{\mathbf{e}}, \overline{\mathbf{e}}_\perp]\mathbf{b}$, where $\mathbf{b} \in \mathbb{R}^{2\times 1}$ is an eigenvector of the $2 \times 2$ matrix:

$$\mathbf{A} := \begin{bmatrix} (1 - \mu^2) & -\rho\mu^2 \\ -\rho\mu^2 & \rho^2(1 - \mu^2) \end{bmatrix}. \tag{48}$$

Also, the corresponding eigenvalues of $\mathbf{Q}$ are the corresponding eigenvalues of $\mathbf{A}$. It can be verified that the eigenvalues of $\mathbf{A}$ are:

$$\widehat{\lambda}_1 = \frac{(1 + \rho^2)(1 - \mu^2)}{2} + \sqrt{\frac{(1 - \rho^2)^2(1 - \mu^2)^2}{4} + \rho^2\mu^4}. \tag{49}$$

and

$$\widehat{\lambda}_2 = \frac{(1 + \rho^2)(1 - \mu^2)}{2} - \sqrt{\frac{(1 - \rho^2)^2(1 - \mu^2)^2}{4} + \rho^2\mu^4}. \tag{50}$$

The corresponding eigenvectors of $\mathbf{A}$ are:

$$\widehat{\mathbf{b}}_1 = \frac{1}{\sqrt{b_{1,1}^2 + b_{1,2}^2}}\begin{bmatrix} b_{1,1} \\ b_{1,2} \end{bmatrix} \tag{51}$$

where $b_{1,1} = \frac{(1-\rho^2)(1-\mu^2)}{2} + \sqrt{\frac{(1-\rho^2)^2(1-\mu^2)^2}{4} + \rho^2\mu^4}$ and $b_{1,2} = -\rho\mu^2$, and

$$\widehat{\mathbf{b}}_2 = \frac{1}{\sqrt{b_{2,1}^2 + b_{2,2}^2}} \begin{bmatrix} b_{2,1} \\ b_{2,2} \end{bmatrix}, \tag{52}$$

where $b_{2,1} = \frac{(1-\rho^2)(1-\mu^2)}{2} - \sqrt{\frac{(1-\rho^2)^2(1-\mu^2)^2}{4} + \rho^2\mu^4}$ and $b_{2,2} = -\rho\mu^2$. Thus, the eigenvalues of $\mathbf{Q}$ are $\widehat{\lambda}_1$ and $\widehat{\lambda}_2$; the corresponding eigenvectors are $\widehat{\mathbf{v}}_1 = [\overline{\mathbf{e}}, \overline{\mathbf{e}}_\perp]\widehat{\mathbf{b}}_1$ and $\widehat{\mathbf{v}}_2 = [\overline{\mathbf{e}}, \overline{\mathbf{e}}_\perp]\widehat{\mathbf{b}}_2$. Note that:

$$\frac{(1-\rho^2)(1-\mu^2)}{2} \leq \sqrt{\frac{(1-\rho^2)^2(1-\mu^2)^2}{4} + \rho^2\mu^4} \leq \frac{(1-\rho^2)(1-\mu^2)}{2} + \rho\mu^2.$$

Let us set $\sqrt{\frac{(1-\rho^2)^2(1-\mu^2)^2}{4} + \rho^2\mu^4} = \frac{(1-\rho^2)(1-\mu^2)}{2} + \beta\rho^2\mu^2$, for some $\beta \in (0,1]$. That gives us:

$$\mu = \sqrt{\frac{\beta(1-\rho^2)}{(1+\beta)(1-\beta\rho^2)}}. \tag{53}$$

In that case, we have:

$$\widehat{\lambda}_1 = \frac{1+\beta\rho^2}{1+\beta} \text{ and } \widehat{\lambda}_2 = \rho^2\left(\frac{1-\beta}{1-\beta\rho^2}\right). \tag{54}$$

Also,

$$b_{1,1} = \frac{1-\rho^2}{(1+\beta)(1-\beta\rho^2)}, b_{1,2} = b_{2,2} = -\frac{\beta\rho(1-\rho^2)}{(1+\beta)(1-\beta\rho^2)}, \text{ and } b_{2,1} = -\frac{\beta^2\rho^2(1-\rho^2)}{(1+\beta)(1-\beta\rho^2)}. \tag{55}$$

Therefore,

$$\widehat{\mathbf{b}}_1 = \frac{1}{\sqrt{1+\beta^2\rho^2}} \begin{bmatrix} 1 \\ -\beta\rho \end{bmatrix} \text{ and } \widehat{\mathbf{b}}_2 = \frac{1}{\sqrt{1+\beta^2\rho^2}} \begin{bmatrix} -\beta\rho \\ -1 \end{bmatrix}. \tag{56}$$

Recall that the eigenvalues of $\mathbf{Q}$ are $\widehat{\lambda}_1$ and $\widehat{\lambda}_2$, and the corresponding eigenvectors are

$$\widehat{\mathbf{v}}_1 = [\overline{\mathbf{e}}, \overline{\mathbf{e}}_\perp]\widehat{\mathbf{b}}_1 = \frac{1}{\sqrt{1+\beta^2\rho^2}}\overline{\mathbf{e}} - \frac{\beta\rho}{\sqrt{1+\beta^2\rho^2}}\overline{\mathbf{e}}_\perp \text{ and } \widehat{\mathbf{v}}_2 = [\overline{\mathbf{e}}, \overline{\mathbf{e}}_\perp]\widehat{\mathbf{b}}_2 = -\frac{\beta\rho}{\sqrt{1+\beta^2\rho^2}}\overline{\mathbf{e}} - \frac{1}{\sqrt{1+\beta^2\rho^2}}\overline{\mathbf{e}}_\perp.$$

Finally, recall that $\mu = \sqrt{\frac{\beta(1-\rho^2)}{(1+\beta)(1-\beta\rho^2)}}$. $\qquad\square$

**Lemma F.4.** *Recall that the averaged model with parameter $\omega$ as defined in Equation (16) was*

$$\boldsymbol{\theta}_{\text{avg}}(\omega) = \omega\boldsymbol{\theta}_* + (1-\omega)\widetilde{\boldsymbol{\theta}}_* = \widetilde{\boldsymbol{\theta}}_* + \omega\mathbf{e}.$$

*We have:*

$$\min_{\omega\in[0,1]} \text{err}_{\text{tot}}(\boldsymbol{\theta}_{\text{avg}}(\omega)) = \left(\frac{\overline{\mathbf{e}}^\top\boldsymbol{\Sigma}\overline{\mathbf{e}}}{\overline{\mathbf{e}}^\top\boldsymbol{\Sigma}\overline{\mathbf{e}}+1}\right)\|\mathbf{e}\|_2^2, \tag{57}$$

*where recall that $\boldsymbol{\Sigma}$ is the covariance matrix of the pre-training data.*

*Proof.* We have:

$$\text{err}_{\text{tot}}(\boldsymbol{\theta}_{\text{avg}}(\omega)) = \text{err}_1(\boldsymbol{\theta}_{\text{avg}}(\omega)) + \text{err}_2(\boldsymbol{\theta}_{\text{avg}}(\omega)) = (\boldsymbol{\theta}_{\text{avg}}(\omega)-\boldsymbol{\theta}_*)^\top\boldsymbol{\Sigma}(\boldsymbol{\theta}_{\text{avg}}(\omega)-\boldsymbol{\theta}_*) + (\boldsymbol{\theta}_{\text{avg}}(\omega)-\widetilde{\boldsymbol{\theta}}_*)^\top\widetilde{\boldsymbol{\Sigma}}(\boldsymbol{\theta}_{\text{avg}}(\omega)-\widetilde{\boldsymbol{\theta}}_*). \tag{58}$$

Plugging in the value of $\boldsymbol{\theta}_{\text{avg}}(\omega)$ and using the value of $\widetilde{\boldsymbol{\Sigma}}$ from Equation (3) above, we get:

$$\text{err}_{\text{tot}}(\boldsymbol{\theta}_{\text{avg}}(\omega)) = (1-\omega)^2\mathbf{e}^\top\boldsymbol{\Sigma}\mathbf{e} + \omega^2\|\mathbf{e}\|_2^2. \tag{59}$$

It can be verified (with elementary calculus) that the optimal value of $\omega$ that minimizes the RHS in Equation (59) is $\omega^* = \frac{\mathbf{e}^\top\boldsymbol{\Sigma}\mathbf{e}}{\mathbf{e}^\top\boldsymbol{\Sigma}\mathbf{e}+\|\mathbf{e}\|_2^2}$. Plugging this into Equation (59) and simplifying a bit yields the desired result. $\qquad\square$

**Lemma F.5.** *Suppose $\alpha > 0$ and $\mathbf{r} \in \mathbb{R}^d$ is a unit-norm vector, i.e., $\|\mathbf{r}\|_2 = 1$. Let*

$$\mathbf{M} := \mathbb{E}\left[ \exp\left( -\frac{\left(\langle \mathbf{r}, \mathbf{z}\rangle\right)^2}{\alpha} \right) \mathbf{z}\mathbf{z}^\top \right],$$

*where $\mathbf{z} \sim \mathcal{N}(\vec{0}_d, \mathbf{I}_d)$. $\mathbf{r}$ is an eigenvector of $\mathbf{M}$ with eigenvalue $\left(\frac{\alpha}{\alpha+2}\right)^{3/2}$. Further, the eigenvectors of $\mathbf{M}$ in the subspace of $\mathbb{R}^d$ orthogonal to $\mathbf{r}$ all have eigenvalues $\left(\frac{\alpha}{\alpha+2}\right)^{1/2}$.*

*Proof.* We have:

$$\mathbb{E}\left[\mathbf{M}\mathbf{r}\right] = \mathbb{E}\left[ \exp\left( -\frac{\left(\langle \mathbf{r}, \mathbf{z}\rangle\right)^2}{\alpha} \right) \langle \mathbf{r}, \mathbf{z}\rangle \mathbf{z} \right]. \tag{60}$$

Suppose $\{\mathbf{r}_{\perp,j}\}_{j=1}^{d-1}$ is an orthonormal basis for the subspace orthogonal to $\mathbf{r}$; so $\langle \mathbf{r}_{\perp,j}, \mathbf{r}\rangle = 0 \ \forall \ j \in [d-1]$ and $\langle \mathbf{r}_{\perp,j}, \mathbf{r}_{\perp,k}\rangle = \mathbb{1}(j = k) \ \forall \ j, k \in [d-1]$. Then, note that:

$$\mathbf{z} = \langle \mathbf{r}, \mathbf{z}\rangle \mathbf{r} + \sum_{j=1}^{d-1} \langle \mathbf{r}_{\perp,j}, \mathbf{z}\rangle \mathbf{r}_{\perp,j}. \tag{61}$$

Since $\mathbf{z} \sim \mathcal{N}(\vec{0}_d, \mathbf{I}_d)$, $\langle \mathbf{r}, \mathbf{z}\rangle$ and $\{\langle \mathbf{r}_{\perp,j}, \mathbf{z}\rangle\}_{j=1}^{d-1}$ are i.i.d. $\mathcal{N}(0, 1)$. Using all of this in Equation (60), we get:

$$\mathbb{E}\left[\mathbf{M}\mathbf{r}\right] = \mathbb{E}\left[ \exp\left( -\frac{\left(\langle \mathbf{r}, \mathbf{z}\rangle\right)^2}{\alpha} \right) \left(\langle \mathbf{r}, \mathbf{z}\rangle\right)^2 \right] \mathbf{r} + \sum_{j=1}^{d-1} \underbrace{\mathbb{E}\left[ \exp\left( -\frac{\left(\langle \mathbf{r}, \mathbf{z}\rangle\right)^2}{\alpha} \right) \langle \mathbf{r}, \mathbf{z}\rangle \langle \mathbf{r}_{\perp,j}, \mathbf{z}\rangle \right]}_{=0 \ (\langle \mathbf{r}, \mathbf{z}\rangle \text{ and } \langle \mathbf{r}_{\perp,j}, \mathbf{z}\rangle \text{ are independent})} \mathbf{r}_{\perp,j} \tag{62}$$

$$= \mathbb{E}_{Z \sim \mathcal{N}(0,1)}\left[ \exp\left( -\frac{Z^2}{\alpha} \right) Z^2 \right] \mathbf{r} \qquad \text{(because } \langle \mathbf{r}, \mathbf{z}\rangle \sim \mathcal{N}(0, 1)\text{)} \tag{63}$$

$$= \left( \frac{1}{\sqrt{2\pi}} \int_{-\infty}^{\infty} z^2 \exp\left( -z^2\left(\frac{1}{\alpha} + \frac{1}{2}\right)\right) dz \right) \mathbf{r} \tag{64}$$

$$= \left( \frac{\alpha}{\alpha + 2} \right)^{3/2} \mathbf{r}. \tag{65}$$

So $\mathbf{r}$ is an eigenvector of $\mathbf{M}$ with eigenvalue $\left(\frac{\alpha}{\alpha+2}\right)^{3/2}$.

Next, note that:

$$\mathbb{E}\left[\mathbf{M}\mathbf{r}_{\perp,1}\right] = \underbrace{\mathbb{E}\left[ \exp\left( -\frac{\left(\langle \mathbf{r}, \mathbf{z}\rangle\right)^2}{\alpha} \right) \langle \mathbf{r}_{\perp,1}, \mathbf{z}\rangle \langle \mathbf{r}, \mathbf{z}\rangle \right]}_{=0} \mathbf{r} + \mathbb{E}\left[ \exp\left( -\frac{\left(\langle \mathbf{r}, \mathbf{z}\rangle\right)^2}{\alpha} \right) \left(\langle \mathbf{r}_{\perp,1}, \mathbf{z}\rangle\right)^2 \right] \mathbf{r}_{\perp,1}$$

$$\sum_{j=2}^{d-1} \underbrace{\mathbb{E}\left[ \exp\left( -\frac{\left(\langle \mathbf{r}, \mathbf{z}\rangle\right)^2}{\alpha} \right) \langle \mathbf{r}_{\perp,1}, \mathbf{z}\rangle \langle \mathbf{r}_{\perp,j}, \mathbf{z}\rangle \right]}_{=0} \mathbf{r}_{\perp,j}. \tag{66}$$

In the above equation, the first and last terms are 0 because $\langle \mathbf{r}, \mathbf{z}\rangle$ and $\{\langle \mathbf{r}_{\perp,j}, \mathbf{z}\rangle\}_{j=1}^{d-1}$ are i.i.d. $\mathcal{N}(0, 1)$; using this fact again,

we get:

$$\mathbb{E}\Big[\mathbf{M}\mathbf{r}_{\perp,1}\Big] = \mathbb{E}_{Z\sim\mathcal{N}(0,1)}\bigg[\exp\Big(-\frac{Z^2}{\alpha}\Big)\bigg]\underbrace{\mathbb{E}_{\bar{Z}\sim\mathcal{N}(0,1)}\big[\bar{Z}^2\big]}_{=1}\mathbf{r}_{\perp,1} \tag{67}$$

$$= \bigg(\frac{1}{\sqrt{2\pi}}\int_{-\infty}^{\infty}\exp\Big(-z^2\Big(\frac{1}{\alpha}+\frac{1}{2}\Big)\Big)dz\bigg)\mathbf{r}_{\perp,1} \tag{68}$$

$$= \Big(\frac{\alpha}{\alpha+2}\Big)^{1/2}\mathbf{r}_{\perp,1}. \tag{69}$$

Similarly, we can show that for $j = \{2,\ldots,d-1\}$, we have:

$$\mathbb{E}\Big[\mathbf{M}\mathbf{r}_{\perp,j}\Big] = \Big(\frac{\alpha}{\alpha+2}\Big)^{1/2}\mathbf{r}_{\perp,j}. \tag{70}$$

So for all $j \in [d-1]$, $\mathbf{r}_{\perp,j}$ is an eigenvector of $\mathbf{M}$ with eigenvalue $\big(\frac{\alpha}{\alpha+2}\big)^{1/2}$. Thus, the eigenvectors of $\mathbf{M}$ in the subspace orthogonal to $\mathbf{r}$ all have eigenvalues $\big(\frac{\alpha}{\alpha+2}\big)^{1/2}$. $\qquad\square$

# G. Experimental Details

In this section, we further discuss the experimental setup of FLOW's usage in both language and vision settings, specifically covering the following:

- Appendix G.1: Baseline Details.

- Appendix G.2: Language Model Hyper-Parameters.

- Appendix G.3: Language Model Evaluation Details.

- Appendix G.4: Vision Model Implementation Details.

### G.1. Baseline Details

In this section, we further discuss the baselines mentioned in Section 5.

**Linear Probing:**  In our vision experiments, we define linear probing as freezing the body of the pre-trained model, initializing a new (task-specific) head and batch normalization layers, and training only the new head and batch normalization layers.

$\ell_2$ **regularization:**  Based on Kirkpatrick et al. (2016), we perform $\ell_2$ regularization as a baseline in the data-oblivious setting. Specifically, the $\ell_2$-regularized loss is:

$$\mathcal{L}(\boldsymbol{\theta}) = \sum_{i=1}^{n} f_i(\boldsymbol{\theta}) + \lambda\|\boldsymbol{\theta}-\boldsymbol{\theta}^*\|_2^2 \tag{71}$$

where $f_i$ is the $i^{\text{th}}$ sample's loss, $\boldsymbol{\theta}^*$ is the pre-trained model, and $\lambda$ is the regularization parameter. Intuitively, as $\lambda$ increases, our model stays closer to the pre-trained model, mitigating forgetting at the expense of target domain performance.

**LoRA (Hu et al., 2022):**  Recently, Biderman et al. (2024) showed that fine-tuning language models with LoRA (Hu et al., 2022) effectively mitigates forgetting. Following a similar setup as us, Biderman et al. (2024) fine-tuned language models on MetaMathQA (Yu et al., 2023) and then evaluated the fine-tuned model on several general capability tasks, viz., HellaSwag (Zellers et al., 2019), ARC-c (Clark et al., 2018), and WinoGrande (Sakaguchi et al., 2019), and one target domain task, viz., GSM8K (Cobbe et al., 2021). Further details about experimental hyper-parameters can be found in Appendix G.2.

**WiSE-FT (Wortsman et al., 2021):** We also consider model averaging as a baseline, specifically focusing on WiSE-FT (Wortsman et al., 2021). WiSE-FT is simply the convex combination of the model parameters shared between the two tasks, while the task-specific parts are not averaged. Specifically, we perform model averaging between the pre-trained model and the fine-tuned model. The convex combination parameter $\alpha$ of WiSE-FT is set to $0.5$ in our experiments, as we cannot optimize $\alpha$ in the data-oblivious setting.

**Learning without Forgetting (Li & Hoiem, 2016):** This work considers using a distillation-based loss to mitigate forgetting when the data from the training of previous tasks is not available. Initially, they record the responses $y_o$ on the new task images with the old tasks' model parameters, then train the model using a combination of fine-tuning loss, distillation loss, and model regularization. For new tasks, they use standard cross-entropy loss, while for old tasks they employ distillation loss that encourages the updated model's responses to match the recorded responses $y_o$ of the original model. Their proposed method has a distillation loss scaling factor $\lambda_0$ and a temperature parameter $T$ in the distillation loss, introducing extra tunable parameters. Their loss induces joint optimization of shared parameters $\theta_s$, old task parameters $\theta_o$, and new task parameters $\theta_n$ using only new task data. Note that unlike FLOW, they also update the old task parameters $\theta_o$.

### G.2. Language Model Hyper-Parameters

For both Gemma 2 2B (Team et al., 2024) and Llama 3.2 3B (Grattafiori et al., 2024), we run hyper-parameter sweeps on learning rates for each baseline. For standard fine-tuning, $\ell_2$ regularization, and FLOW, we do a learning rate sweep in [1e-4, 2e-5, 1e-5, 5e-6], and for LoRA ($r = 64$) we do a sweep in [2e-4, 2e-1], following the learning rates used in (Biderman et al., 2024). We then select the learning rate that results in the best GSM8K (Cobbe et al., 2021) accuracy, oblivious to general capability metrics. We report the hyper-parameters used for our Gemma 2 2B (Team et al., 2024) experiments in Table 6 and for Llama 3.2 3B (Grattafiori et al., 2024) in Table 7.

*Table 6.* The hyper-parameters used to train Gemma 2 2B in our experiments. Note that the learning rate selected is based on the best results on GSM8K after fine-tuning the method on MetaMathQA.

| Hyper-parameter | Standard Fine-tuning | LoRA ($r = 64$) | $\ell_2$-Reg. | FLOW (Ours) |
|---|---|---|---|---|
| Learning Rate | 1e-5 | 2e-4 | 5e-6 | 5e-6 |
| Learning Rate Scheduler | | Cosine | | |
| Batch Size | | 128 | | |
| Optimizer | | AdamW | | |
| Weight Decay | | 0.00 | | |
| Warmup Ratio | | 0.03 | | |
| Epochs | | 2 | | |
| Max Sequence Length | | 1024 | | |
| Seed | | 42 | | |

*Table 7.* The hyper-parameters used to train Llama 3.2 3B in our experiments. Note that the learning rate selected is based on the best results on GSM8K after fine-tuning the method on MetaMathQA.

| Hyper-parameter | Standard Fine-tuning | LoRA ($r = 64$) | $\ell_2$-Reg. | FLOW (Ours) |
|---|---|---|---|---|
| Learning Rate | 2e-5 | 2e-4 | 1e-5 | 1e-5 |
| Learning Rate Scheduler | | Cosine | | |
| Batch Size | | 128 | | |
| Optimizer | | AdamW | | |
| Weight Decay | | 0.00 | | |
| Warmup Ratio | | 0.03 | | |
| Epochs | | 2 | | |
| Max Sequence Length | | 1024 | | |
| Seed | | 42 | | |

For our WiSE-FT (Wortsman et al., 2021) model averaging experiments, we use $\alpha = 0.5$. For our LoRA (Hu et al., 2022) experiments, we use $\alpha = r = 64$. For $\ell_2$ regularization we use $\lambda = 1e - 3$ which is taken from (Chen et al., 2024b). Most training hyper-parameters for our language experiments are taken from Chen et al. (2024b), with the introduction of learning rate sweeps.

### G.3. Language Model Evaluation Details

As described in Section 5.2, we create a commonsense reasoning metric composed of the following six metrics: ARC-e (Clark et al., 2018), ARC-c (Clark et al., 2018), HellaSwag (Zellers et al., 2019), PIQA (Bisk et al., 2020), SIQA (Sap et al., 2019), and OBQA (Mihaylov et al., 2018). On top of the commonsense metric, we evaluate MMLU (Hendrycks et al., 2021a) and MBPP (Austin et al., 2021) to estimate the general capabilities of a language model and to measure the effects of catastrophic forgetting when fine-tuning a model on MetaMathQA (Yu et al., 2023). We additionally use GSM8K (Cobbe et al., 2021) to evaluate the target fine-tuning performance of a given fine-tuning method. We provide a brief describe each of these evaluation metrics:

1. **HellaSwag** (Zellers et al., 2019): A benchmark designed to test commonsense reasoning. HellaSwag presents a context followed by several plausible endings, and the model must choose the most appropriate continuation.

2. **ARC Easy** (Clark et al., 2018): A benchmark part of the AI2 reasoning challenge designed to test basic scientific reasoning and knowledge. ARC Easy presents 5,197 multiple-choice science questions drawn from grade 3-9 standardized tests, where each question typically includes a brief scientific scenario or statement followed by four possible answer choices.

3. **ARC Challenge** (Clark et al., 2018): A benchmark part of the AI2 reasoning challenge designed to test advanced scientific reasoning and knowledge application. ARC Challenge presents 2,590 multiple-choice science questions drawn from grade 3-9 standardized tests, where each question typically includes a scientific scenario or phenomenon followed by four possible answer choices. The questions in ARC Challenge are significantly more challenging than ARC Easy.

4. **PIQA** (Bisk et al., 2020): A benchmark designed to evaluate physical commonsense understanding in natural language. PIQA presents a goal and two possible solutions, requiring models to choose the most appropriate solution that demonstrates an understanding of everyday physical interactions.

5. **SIQA** (Sap et al., 2019): A benchmark designed to evaluate social commonsense intelligence and emotional reasoning. SIQA presents a social situation context followed by a question and three possible answers, requiring models to demonstrate an understanding of social interactions, emotional responses, and behavioral implications.

6. **Open Book QA** (Mihaylov et al., 2018): A benchmark designed to assess understanding of elementary science concepts in an open-book exam format. OBQA presents 5,957 multiple-choice questions paired with a small "book" of 1,326 core science facts, requiring models to combine these facts with common knowledge to arrive at correct answers.

7. **MMLU** (Hendrycks et al., 2021a): A benchmark designed to evaluate massive multitask language understanding. MMLU presents approximately 16,000 multiple-choice questions spanning 57 subjects including mathematics, philosophy, law, and medicine, requiring models to demonstrate broad knowledge and reasoning capabilities.

8. **MBPP** (Austin et al., 2021): A benchmark designed to evaluate basic Python programming capabilities. The entire MBPP dataset presents 974 Python programming problems, where each problem includes a natural language task description and three test cases written as assert statements, requiring models to generate functionally correct Python code solutions.

9. **GSM8K** (Cobbe et al., 2021): A benchmark designed to evaluate multi-step mathematical reasoning capabilities. The GSM8K test set contains 1,000 grade school math word problems, where each problem requires 2-8 steps to solve using basic arithmetic operations (addition, subtraction, multiplication, division).

We follow the standard evaluation process for each of these datasets and specifically use `lm-evaluation-harness` (Gao et al., 2024) to evaluate our experiments.

## G.4. Vision Model Implementation Details

We performed an extensive hyper-parameter search over six learning rates (lrs = $[0.05, 0.01, 0.005, 0.001, 0.0005, 0.0001]$), two models, and six datasets (i.e., 72 total runs per method) for standard fine-tuning, linear probing, and FLOW. We chose the best learning rates associated with the highest average score over all the target (fine-tuning) datasets. Since our method is data oblivious, we do not use the validation set of ImageNet-1K other than for evaluation. For training models with $\ell_2$-regularization, we adapted the same learning rates and other related hyperparameters used for standard fine-tuning. We searched for $\lambda$ using one dataset and ResNet50 model ($\lambda = [0.002, 0.00001, 0.00002]$) and chose 0.002 based on average accuracy over target data. We chose ($\alpha = 0.05$) for WiSE-FT following Wortsman et al. (2021). We present all the important training details in Table 8 for the ResNet models.

*Table 8.* Hyperparameter configurations for finetuning ResNet18 and ResNet50 on the image classification datasets.

| Model | Hyperparameters | CIFAR10 | CIFAR100 | Flowers102 | Caltech101 | Dogs | Cars |
|---|---|---|---|---|---|---|---|
| | # GPUs | | | 1 A6000 | | | |
| | Optimizer | | | SGD | | | |
| | LR Schedule | | | Cosine (except for Linear probing) | | | |
| | Weight Decay | | | 0.0005 | | | |
| | Seed | | | 42 | | | |
| | $\lambda$ for $\ell_2$-Reg. | | | 0.002 | | | |
| | $\alpha$ for WiSE-FT. | | | 0.05 | | | |
| | $\tau$ (temperature) for FLOW | | | median loss | | | |
| | Epochs | 20 | 25 | 25 | 30 | 30 | 30 |
| ResNet18 | LR-Standard fine-tuning | 5E-3 | 1E-2 | 5E-2 | 5E-3 | 1E-3 | 5E-2 |
| | LR-Linear probing | 5E-3 | 5E-3 | 5E-2 | 1E-2 | 5E-3 | 5E-2 |
| | LR-FLOW | 1E-3 | 5E-3 | 5E-2 | 1E-2 | 5E-3 | 1E-2 |
| ResNet50 | LR-Standard fine-tuning | 5E-3 | 1E-3 | 1E-2 | 5E-3 | 5E-4 | 5E-2 |
| | LR-Linear probing | 5E-2 | 5E-2 | 5E-2 | 5E-2 | 1E-2 | 5E-2 |
| | LR-FLOW | 5E-4 | 1E-3 | 1E-2 | 1E-2 | 5E-3 | 1E-2 |

**Hyperparameter Details for ViT-B/16.** We fine-tuned for 8 epochs on Food-101. We used a learning rate of 5e–5 for standard fine-tuning, $\ell_2$-regularization (with $\lambda = 0.002$), and our method FLOW. For linear probing, we used a higher learning rate of 1e–3. For LwF, we used the same learning rate as standard fine-tuning (5e–5), $T = 2$, and $\lambda_0 = 5$e–4. Here we used a higher value of $\tau$ for FLOW and set it equal to $80^{\text{th}}$ percentile of the pre-trained loss values.

**Datasets**

1. **ImageNet-1K** (Russakovsky et al., 2015) serves as the pre-training dataset for all our vision base models. It is a widely used large-scale image classification dataset, consisting of over a million images spanning 1000 classes.

2. **CIFAR-10** (Krizhevsky, 2009) is a widely used dataset for image classification tasks. It consists of 60,000 32x32 color images divided into ten classes, with 6,000 images per class.

3. **CIFAR-100** (Krizhevsky, 2009) extends CIFAR-10 by providing 100 classes containing 600 images each. This dataset is used for fine-grained image classification tasks.

4. **Caltech101** (Li et al., 2022) comprises images of a diverse range of objects across 101 categories with diverse set of image classes.

5. **Flowers102** (Nilsback & Zisserman, 2008) comprises 102 categories of flowers, with each category containing between 40 to 258 images. This dataset is commonly used for fine-grained image classification and flower recognition tasks.

6. **Cars** (Krause et al., 2013) refers to the Stanford Cars dataset, which includes 16,185 images of 196 classes of cars. It provides a rich resource for fine-grained car classification task.

7. **Dogs** (Parkhi et al., 2012) pertains to the Stanford Dogs dataset, containing 20,580 images of 120 breeds of dogs. This dataset is widely used for fine-grained dog breed classification and recognition tasks.

8. **Food101** (Bossard et al., 2014) is a large-scale dataset for food classification containing 101 categories with 1,000 images per class, commonly used to evaluate models on fine-grained object recognition tasks.

## H. Detailed Vision Results and Ablations

First, we present the detailed version of the results of Table 1 in Tables 9 and 10.

Table 9. **ResNets: Target accuracies on each of the six datasets for the results in Table 1.**

| | Method | CIFAR-10 | CIFAR-100 | Flowers-102 | Caltech-101 | Dogs | Cars | Average |
|---|---|---|---|---|---|---|---|---|
| ResNet18 | Linear probing | 81.32 | 60.06 | 87.20 | 91.15 | 78.50 | 43.23 | 73.57 |
| | Standard FT | 96.15 | 83.42 | 92.45 | 94.02 | 80.47 | 87.91 | 89.07 |
| | $\ell_2$-Regularization | 95.53 | 81.82 | 92.11 | 94.23 | 80.27 | 84.78 | 88.12 |
| | WiSE-FT | 91.47 | 65.90 | 87.28 | 91.40 | 82.48 | 62.88 | 80.23 |
| | **FLOW** (Ours) | 88.25 | 78.95 | 90.01 | 93.05 | 86.20 | 67.17 | 83.93 |
| ResNet50 | Linear probing | 86.62 | 67.80 | 83.64 | 93.45 | 85.76 | 41.97 | 76.45 |
| | Standard FT | 97.61 | 86.11 | 91.74 | 96.02 | 89.26 | 89.94 | 91.78 |
| | $\ell_2$-Regularization | 97.50 | 85.77 | 91.67 | 95.85 | 89.29 | 89.42 | 91.58 |
| | WiSE-FT | 94.65 | 72.55 | 71.95 | 93.73 | 92.52 | 62.89 | 81.38 |
| | **FLOW** (Ours) | 91.11 | 79.42 | 86.78 | 94.45 | 91.16 | 74.59 | 86.25 |

Table 10. **ResNets: Top-1 ImageNet-1K accuracy after fine-tuning on each target dataset for the results in Table 1.**

| | Method | CIFAR-10 | CIFAR-100 | Flowers-102 | Caltech-101 | Dogs | Cars | Average |
|---|---|---|---|---|---|---|---|---|
| ResNet18 | Linear probing | 69.76 | 69.76 | 69.76 | 69.76 | 69.76 | 69.76 | 69.76 |
| | Standard FT | 19.93 | 0.39 | 6.48 | 34.17 | 56.38 | 0.17 | 19.58 |
| | $\ell_2$-Regularization | 37.86 | 29.86 | 19.34 | 46.67 | 58.34 | 16.64 | 34.78 |
| | WiSE-FT | 62.24 | 47.65 | 49.98 | 64.70 | 67.34 | 33.03 | 54.15 |
| | **FLOW** (Ours) | 69.02 | 52.64 | 67.80 | 68.32 | 67.78 | 65.74 | 65.21 |
| ResNet50 | Linear probing | 79.02 | 79.02 | 79.02 | 79.02 | 79.02 | 79.02 | 79.02 |
| | Standard FT | 16.89 | 35.95 | 61.01 | 40.51 | 66.93 | 0.21 | 36.91 |
| | $\ell_2$-Regularization | 33.98 | 47.16 | 62.85 | 43.42 | 67.03 | 14.27 | 44.78 |
| | WiseFT ($\alpha = 0.5$) | 61.40 | 73.04 | 76.33 | 73.25 | 77.36 | 8.55 | 61.65 |
| | **FLOW** (Ours) | 78.26 | 75.13 | 78.60 | 73.38 | 78.55 | 72.64 | 76.09 |

In Table 11, we present a small ablation to compare the pre-training and fine-tuning performances with different values of $\tau$ in FLOW; recall that we prescribed selecting $\tau$ to be the median pre-training loss value. As we see, if we could tune $\tau$, then FLOW's results would be even better.

Table 11. FLOW **ablation with different values of** $\tau$: Pre-training and fine-tuning accuracies as a function of $\tau$ set to different percentiles of the pre-training losses. The model is ResNet-50, pre-training dataset is ImageNet-1K (IN-1K), and the fine-tuning dataset is Caltech101 (Target). So if we could tune $\tau$, then FLOW's results would further improve.

| IN-1K Accuracy | Target Accuracy | Average | $\tau$- Percentile (%) |
|---|---|---|---|
| 68.51 | 91.15 | 79.83 | 10 |
| 64.13 | 91.01 | 77.57 | 30 |
| 54.72 | 91.80 | 73.26 | 50 |
| 45.59 | 92.91 | 69.25 | 70 |
| 20.51 | 94.02 | 57.27 | 90 |

Further, in Figure 2, we present another ablation study where we compare FLOW with different values of $\tau$, WiSE-FT with different values of the convex combination parameter, and random selection where we train on a random subset of the fine-tuning data to limit the drift from the pre-trained model.

### H.1. Comparison with a Distillation-Based Method for Mitigating Forgetting

Here, we compare our method FLOW against a distillation-based method for mitigating forgetting called "learning without forgetting" (LwF) (Li & Hoiem, 2016) in vision. More details about this method can be found in Appendix G.1, but

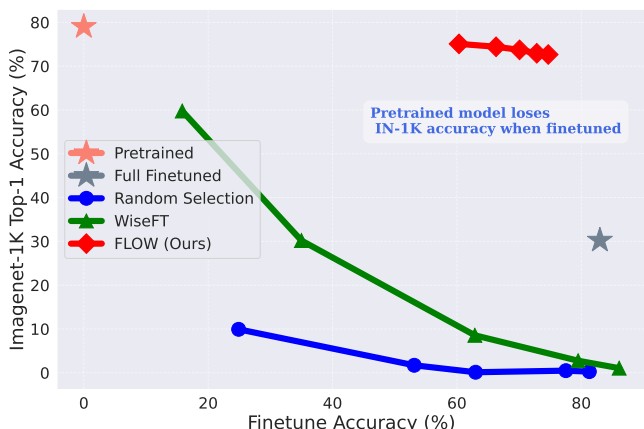

*Figure 2.* **Comparison of FLOW with different values of $\tau$ and some other baselines with different hyper-parameter values.** This plot is for ResNet-50 on the Stanford cars dataset. FLOW's plot (in red) is with $\tau = \{10, 20, 30, 40, 50\}$ percentile of the per-sample losses. As the name "random selection" implies, we just pick a random subset of the fine-tuning data and train on this subset to limit the drift from the pre-trained model. To have some correspondence with our choice of $\tau$ for FLOW, we pick random $\{10, 20, 30, 40, 50\}$ % of the data in "random selection". As we see, FLOW significantly outperforms other methods.

the important thing to note is that it updates the head corresponding to the pre-training data (which FLOW does not do) and also comes with more tunable parameters, additional memory, and evaluation overhead compared to FLOW. We consider the ViT-B/16 (Dosovitskiy et al., 2021) model pre-trained on Imagenet-1K (IN-1K) and a single large fine-tuning dataset, Food101 (Bossard et al., 2014). The evaluation metric is the same as in Section 5.1. Hyper-parameter details are in Appendix G.4.

In Table 12, we list the accuracies of the pre-trained model, standard FT, linear probing, $\ell_2$-regularization, LwF, and FLOW. Note that FLOW outperforms LwF despite its simplicity. Specifically, FLOW achieves better performance on the forgetting front, while LwF does better on the fine-tuning task.

*Table 12.* **Comparison with the distillation-based method (LwF) of Li & Hoiem (2016).** **Bolded** and underlined values indicate the **best** and second-best accuracies within each column (and for each model). Deltas (in color) for IN-1K and target performance are computed w.r.t. the pre-trained and standard fine-tuned models. Note that **FLOW outperforms LwF** despite not updating the head for the pre-training data, unlike LwF, and being more efficient than LwF, which comes with more tunable parameters, additional memory, and evaluation overhead.

| | Method | IN-1K Accuracy | | Target Accuracy | | Average |
|---|---|---|---|---|---|---|
| ViT-B/16 | Pre-trained | **81.10** | (+0.00) | – | | – |
| | Standard FT | 56.11 | (-24.99) | 91.60 | (+0.00) | 73.86 |
| | Linear Probe | **81.10** | (+0.00) | 83.86 | (-7.74) | 82.48 |
| | $\ell_2$-Reg. | 59.18 | (-21.92) | **91.66** | (+0.06) | 75.42 |
| | LwF | 76.39 | (-4.71) | 91.23 | (-0.37) | 83.81 |
| | **FLOW** (Ours) | 77.94 | (-3.16) | 90.57 | (-1.03) | **84.26** |

# I. Additional Language Model Results and Ablations

In this section, we discuss expanded results and further ablations of FLOW within our language experiments, specifically covering the following:

- Appendix I.1: An expanded table of results on commonsense reasoning tasks along with other baselines.

- Appendix I.2: An additional ablation on *token*-wise weighting scheme for fine-tuning with language data.

- Appendix I.3: An expanded set of plots and results for the combination of FLOW with weight averaging techniques such as Wise-FT (Wortsman et al., 2021).

## I.1. Extended Commonsense Reasoning Results

As discussed in Section 5.2 and Appendix G.3, we evaluate FLOW and the other baselines on various commonsense reasoning tasks within fine-tuning with the procedure described in Section 5.2. We include the exact results of these evaluation metrics for various baselines and FLOW in Table 13. We also include the results of commonsense reasoning metrics for the ablation combining FLOW with LoRA and $\ell_2$-regularization in Table 14.

*Table 13.* **Extended commonsense reasoning metrics for FLOW and other baselines within language modeling**. The performance on commonsense reasoning evaluations when fine-tuning Gemma 2 2B (Team et al., 2024) and Llama 3.2 3B (Grattafiori et al., 2024) on MetaMathQA (Yu et al., 2023). We include the target domain evaluation GSM8K (Cobbe et al., 2021) for convenience. The results show that FLOW can effectively mitigate catastrophic forgetting while still getting strong performance on our target fine-tuning task.

| | Method | ARC-e | ARC-c | HellaSwag | PIQA | SIQA | OBQA | Average | GSM8K |
|---|---|---|---|---|---|---|---|---|---|
| Gemma 2 2B | Pre-trained | 80.18 | 46.84 | 54.95 | 78.67 | 51.33 | 31.40 | 57.23 | 24.49 |
| | Standard Fine-tuning | 76.09 | 42.07 | 54.41 | 76.99 | 48.06 | 32.00 | 55.07 | 63.38 |
| | WiSE-FT | 79.55 | 46.42 | 56.43 | 78.24 | 51.08 | 32.00 | 57.28 | 53.30 |
| | LoRA ($r = 64$) | 77.78 | 44.37 | 54.59 | 76.99 | 50.51 | 29.80 | 55.67 | 60.43 |
| | $\ell_2$-Regularization | 79.08 | 45.99 | 56.21 | 77.20 | 50.97 | 32.60 | 57.01 | 62.85 |
| | **FLOW** (Ours) | 79.76 | 47.18 | 56.23 | 77.69 | 51.48 | 33.20 | 57.59 | 62.55 |
| Llama 3.2 3B | Pre-trained | 74.54 | 42.15 | 55.31 | 76.66 | 47.03 | 31.20 | 54.48 | 26.01 |
| | Standard Fine-tuning | 70.03 | 34.22 | 52.02 | 74.16 | 45.24 | 28.40 | 50.68 | 66.95 |
| | WiSE-FT | 75.63 | 40.79 | 55.18 | 76.93 | 47.34 | 31.40 | 54.54 | 57.01 |
| | LoRA ($r = 64$) | 71.38 | 37.88 | 55.01 | 76.55 | 47.39 | 30.40 | 53.10 | 63.84 |
| | $\ell_2$-Regularization | 73.57 | 38.91 | 54.939 | 76.12 | 47.24 | 30.80 | 53.60 | 66.87 |
| | **FLOW** (Ours) | 74.96 | 39.68 | 55.39 | 76.01 | 47.80 | 32.00 | 54.30 | 65.58 |

*Table 14.* **Extended commonsense reasoning metrics for combining FLOW with other baselines**. The performance on commonsense reasoning evaluations when fine-tuning Gemma 2 2B (Team et al., 2024) baselines in conjunction with FLOW on MetaMathQA (Yu et al., 2023). We include the target domain evaluation GSM8K (Cobbe et al., 2021) for convenience. The results show that FLOW can effectively be used in conjunction with other methods that mitigate catastrophic forgetting.

| Method | ARC-e | ARC-c | HellaSwag | PIQA | SIQA | OBQA | Average | GSM8K |
|---|---|---|---|---|---|---|---|---|
| LoRA ($r = 64$) | 77.78 | 44.37 | 54.59 | 76.99 | 50.51 | 29.80 | 55.67 | 60.43 |
| LoRA ($r = 64$) + **FLOW** | 79.50 | 45.39 | 55.27 | 77.31 | 51.18 | 31.80 | 56.74 | 61.49 |
| $\ell_2$-Regularization | 79.08 | 45.99 | 56.21 | 77.20 | 50.97 | 32.60 | 57.01 | 62.85 |
| $\ell_2$-Regularization + **FLOW** | 79.67 | 47.10 | 56.38 | 77.48 | 51.13 | 33.40 | 57.53 | 62.02 |

Table 13 shows a clear trend that FLOW, can strongly mitigate catastrophic forgetting in comparison to standard fine-tuning. For Gemma 2 2B (Team et al., 2024), we can see that FLOW only has $\sim 0.8\%$ reduction in the performance of the target fine-tuning while on average maintaining the commonsense reasoning abilities of the pre-trained model, a $\sim 2.52\%$ increase over standard fine-tuning. For Llama 3.2 3B (Grattafiori et al., 2024), we can see that FLOW can again maintain the commonsense reasoning abilities of the base pre-trained model while only having a $\sim 1.4\%$ drop on target fine-tuning performance. Overall, FLOW strikes a strong balance between general capabilities and target fine-tuning performance compared to other baselines.

For experiments with Gemma 2 2B (Team et al., 2024), FLOW can on average maintain the best scores on commonsense reasoning tasks. Performing only $\sim 0.8\%$ and $\sim 0.3\%$ worse on GSM8K (Cobbe et al., 2021) in comparison to standard fine-tuning and $\ell_2$ regularization, FLOW can improve on commonsense reasoning metrics by $\sim 2.42\%$ and $\sim 0.58\%$ respectively. Interestingly, in our Llama 3.2 3B (Grattafiori et al., 2024) experiments, we found that WiSE-FT (Wortsman et al., 2021) performed the strongest in preventing catastrophic forgetting of commonsense capabilities ($+0.04$ over the pre-trained model); however, this came at the cost of a significant decrease in GSM8K (Cobbe et al., 2021) accuracy ($-9.94$ under standard fine-tuning). In comparison, FLOW effectively mitigated forgetting in commonsense reasoning metrics ($-0.18$ under the pre-trained model), while achieving significantly higher accuracy in GSM8K (Cobbe et al., 2021) ($-1.37$ under standard fine-tuning).

## I.2. Token-wise Sample Weighting Ablations

In the language experiments, "sample" for FLOW can be defined as an entire sequence or an individual token. The experiments in the main paper treat a sequence as a sample; in that case, the per-sample loss is the average loss over the tokens in the sequence. We call this *sequence*-wise re-weighting. Instead, one could treat a token as a sample in which case the per-sample loss is just the token's loss. We call this *token*-wise re-weighting. We run a small ablation on both *sequence*-wise and *token*-wise re-weighting by following a similar experimental setup as Section 5.2. We train a Gemma 2 2B (Team et al., 2024) on MetaMathQA (Yu et al., 2023) and evaluate it on several general capability and target domain evaluations. The results of this experiment are in Table 15.

*Table 15.* The performance of Gemma 2B 2B on general capabilities metrics compared to target domain performance (GSM8K) when training on MetaMathQA. *Pre-trained* is the base model performance of Gemma 2 2B, *Standard* is the performance after full end-to-end fine-tuning, *Sequence* is our sequence sample weighting schema with FLOW, and *Token* is our token sample weighting schema with FLOW. **Bold** and underlined values indicate the **best** and second-best results respectively within each evaluation metric.

| Method | ARC-e | ARC-c | HellaSwag | PIQA | SIQA | OBQA | MMLU | MBPP | GSM8K |
|---|---|---|---|---|---|---|---|---|---|
| Base | **80.18** | 46.84 | 54.95 | **78.67** | 51.33 | 31.40 | **49.59** | **28.40** | 24.49 |
| Standard | 76.09 | 42.07 | 54.41 | 76.99 | 48.06 | **32.80** | 45.59 | 16.80 | **63.38** |
| Sequence | 79.76 | **47.18** | **56.23** | 77.69 | **51.48** | 33.20 | 49.31 | 26.80 | 62.55 |
| Token | 79.38 | 45.90 | 53.95 | 78.29 | 51.28 | 31.80 | 48.75 | 22.00 | 23.73 |

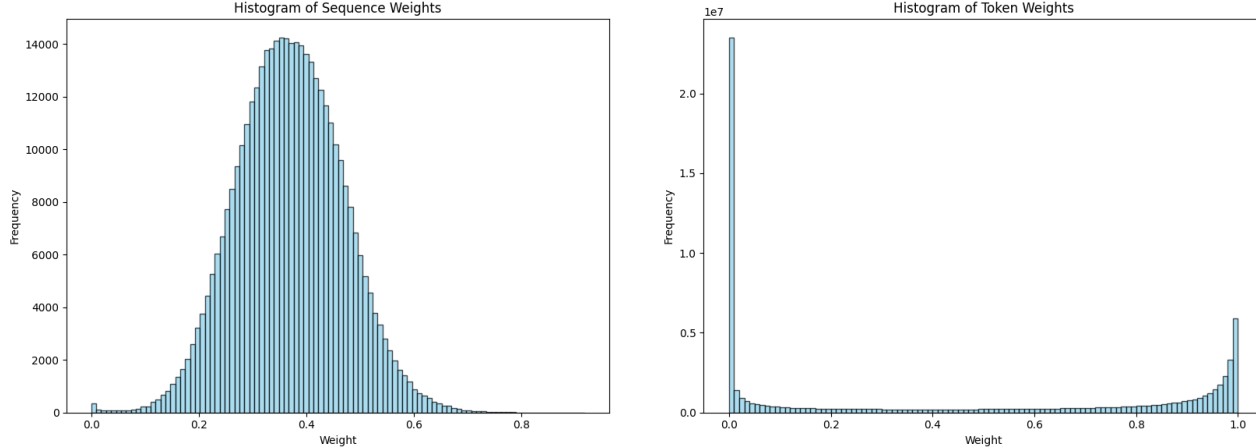

*Figure 3.* Histograms comparing the sample-wise distribution of weights in *sequence*-wise re-weighting schema for FLOW and token-wise distribution of weights *token*-wise re-weighting schema for FLOW. The *sequence*-wise weight distribution is given on the left, while the *token*-wise weight distribution is given on the right.

While *token*-wise sample re-weighting performs comparably or slightly worse than *sequence*-wise sample re-weighting in terms of the catastrophic forgetting of general capabilities of Gemma 2 2B, it struggles to effectively learn the fine-tuning target domain of GSM8K. To further understand this problem, we compare the weight distributions between *sequence*-wise and *token*-wise re-weighting schema in Figure 3. We can see that the *sequence* weights appear Gaussian, while most of the *token* weights are either 0 or 1. We speculate that *token*-wise re-weighting will force any token not commonly appearing in the pre-training data to have a high loss or perplexity, which combined with our algorithm, will heavily down-weight them to almost zero. We further speculate that these tokens are essential to improving the performance of our target fine-tuning task and that using FLOW with a *token*-wise scheme over-regularizes, preventing any meaningful learning of the target task. As *sequence*-wise re-weighting significantly outperforms *token*-wise re-weighting, we recommend using *sequence*-wise re-weighting in FLOW for language models.

## I.3. Extended Weight Averaging Results

As discussed in Section 6, we further combine FLOW with WiSE-FT (Wortsman et al., 2021) to mitigate the effects of catastrophic forgetting when fine-tuning. In this section, we report the full results of combining FLOW and WiSE-FT to prevent catastrophic forgetting with Gemma 2 2B.

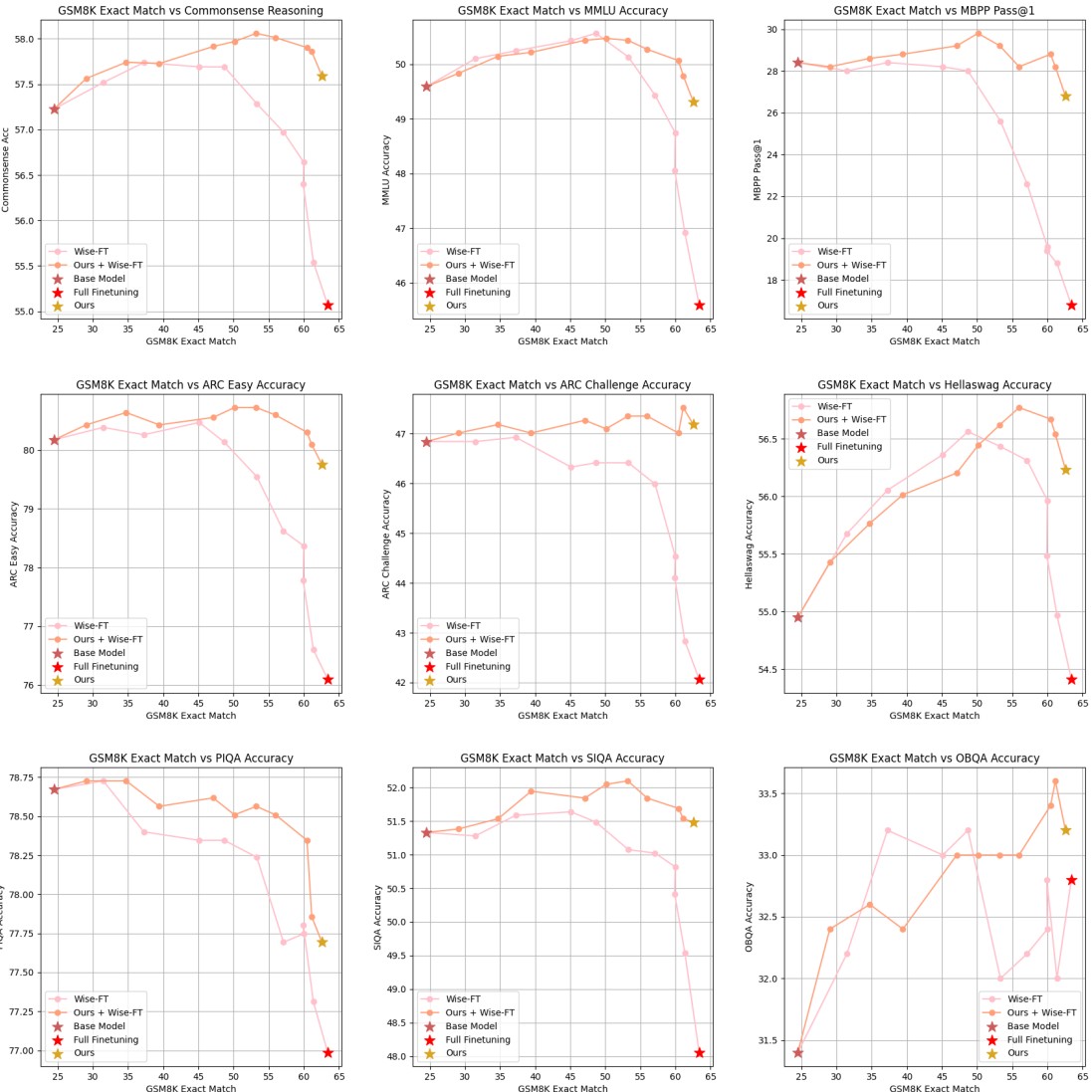

*Figure 4.* **FLOW is complementary with model averaging (WiSE-FT) in language modeling.** We compare WiSE-FT (Wortsman et al., 2021) with a standard model fine-tuning and with FLOW after fine-tuning Gemma 2 2B on MetaMathQA. We use varying $\alpha \in [0, 1]$ for WiSE-FT. The results indicate that combining Wise-FT with FLOW outperforms vanilla WiSE-FT with standard fine-tuning.

