# OpenReview forum: "Upweighting Easy Samples in Fine-Tuning Mitigates Forgetting"
_ICML.cc/2025/Conference — ICML 2025 spotlightposter_

### Official Review · Reviewer_1DND · 2025-03-04

**Overall Recommendation:** 4

**Summary:**

This paper proposes a sample weighting scheme for fine-tuning data based solely on the loss of the pre-trained model.
The proposed method weights samples with low loss of the pre-trained model (i.e., easy samples) to suppress the divergence from the pre-trained model.
Existing methods operate in the parameter or gradient space, but the proposed method is characterised by its emphasis on the sample space.
Furthermore, this paper theoretically analyses the effect of fine-tuning in the linear setting of the proposed method, and shows that learning stops in a specific subspace and that overfitting to the target task is suppressed.
In the experiments, the effectiveness of the proposed method is discussed in both language and visual tasks.

**Claims And Evidence:**

This paper proposes a method for weighting samples with low loss from a pre-trained model to prevent deviation from the pre-trained model.
The effectiveness of this method is discussed from an experimental perspective, but there are some shortcomings in the experiment.
For example, in the experiment on the vision task, only the experiment using ResNet is discussed. In addition, the scale of the dataset is small.
Therefore, it is difficult to judge the effectiveness of the proposed method compared to conventional methods.
On the other hand, the effectiveness of the proposed method is also discussed theoretically. I think that the theoretical analysis is technically sound.

**Essential References Not Discussed:**

The reference meets the minimum required standards.

**Experimental Designs Or Analyses:**

One of the strengths of this paper is that it discusses the effectiveness of the proposed method in both vision tasks and language tasks.
However, as mentioned above, in the experiments for vision tasks, only ResNet was used. In addition, the scale of the dataset is small.
In the experiments, many of the experiments were conducted using small-scale datasets. For example, if a larger dataset than CIFAR is used, will the proposed method work effectively?
In the class identification task, there are many proposals for larger datasets than CIFAR, etc.
The vision model is a small model such as ResNet18 and ResNet50. Can you also argue for the effectiveness of the proposed method in larger transformers such as ViT?
In addition, the method of simple distillation, e.g., [1], is also a standard approach that does not use past data (i.e., data before fine-tuning). Is the proposed method more effective than the method of simple distillation?

[1] Li,Z. andHoiem,D. Learningwithout forgetting. In Leibe,B.,Matas, J.,Sebe,N.,andWelling,M. (eds.) ECCV(4),volume9908ofLectureNotesinComputer Science,pp.614–629.Springer,2016.

**Methods And Evaluation Criteria:**

I think the theoretical analysis of linear regression is technically sound.
However, in order to demonstrate its effectiveness in more practical application scenarios, it is necessary to conduct experiments on large datasets and demonstrate its effectiveness with various models, including ViT.

**Other Comments Or Suggestions:**

n/a

**Other Strengths And Weaknesses:**

This paper proposes a method for preventing deviation from pre-trained models by weighting simple samples with low loss.
The proposed method is simple but effective, and can be applied to various tasks such as image classification tasks and language tasks.
The theoretical analysis is technically sound.
However, there are not enough experiments on image tasks.
Also, the introduction is too long (especially after line 82).
Instead of shortening the introduction, I think that experimental supplements and additions on image tasks are necessary in the main text.

**Questions For Authors:**

In the experiments, many of the experiments were conducted using small-scale datasets. For example, if a larger dataset than CIFAR is used, will the proposed method work effectively?
The vision model is a small model such as ResNet18 and ResNet50. Can you also argue for the effectiveness of the proposed method in larger transformers such as ViT?
In addition, the method of simple distillation, e.g., [1], is also a standard approach that does not use past data (i.e., data before fine-tuning). Is the proposed method more effective than the method of simple distillation?

[1] Li,Z. andHoiem,D. Learningwithout forgetting. In Leibe,B.,Matas, J.,Sebe,N.,andWelling,M. (eds.) ECCV(4),volume9908ofLectureNotesinComputer Science,pp.614–629.Springer,2016.

**Relation To Broader Scientific Literature:**

Yes.
The proposed method and evaluation criteria are appropriate as a problem setting for lifelong learning.
The main contribution of this paper (the upweighting framework) is closely related to several existing methods cited in this paper, as well as general methods such as LoRA.
Experimental comparisons are made with these conventional methods, and the effectiveness of the proposed method compared to conventional methods is discussed.

**Theoretical Claims:**

I think the theoretical analysis in sec.7 and appendix is technically sound.
(However, please note that I have not completely proved all of the mathematical formulas in this paper myself.)

---

> ### Author Rebuttal · Authors · 2025-04-01
>
> We thank the reviewer for their detailed evaluation and encouraging feedback. We respond to your questions and feedback below:
>
> **Performance for larger/different models and larger datasets:** We believe that our language experiments are indeed large-scale, and the size of our vision datasets is comparable to prior works such as [A, B]. However, we appreciate your feedback, and in light of that, we performed extra experiments on larger models (ViT-B/16 and CLIP ViT-B/32) and a larger dataset (Food 101). We provide the details and results below:
>
> * **ViT-B/16 model on Food101:** As requested, we performed experiments with a ViT pre-trained on ImageNet-1k (IN-1K). Also, Food 101 has twice the number of samples (101k samples) as CIFAR (50k samples), the image sizes are very different, and are corrupted with random noise. We follow a similar experimental setup to section 5.1 in the paper. Due to a lack of space, we are omitting the rest of the details here. Here are the results:
>
> | Method                 | IN-1K (top-1) | Food101 | Average |
> |------------------------|---------------|---------|---------|
> | Pre-trained           | 81.10         | ---     | ---     |
> | Standard FT          | 56.11         | 91.60   | 73.86   |
> | Linear probe         | 81.10         | 83.86   | 82.48   |
> | L2 reg              | 59.18         | 91.66   | 75.42   |
> | **FLOW (Ours)**                | 77.94         | 90.57   | **84.26** |
>
> FLOW exhibits the same behavior as in our previous experiments in the paper – *meaningfully outperforming the baselines*.
>
> * **CLIP ViT-B/32:** Here, we fine-tune the image encoder of a pre-trained (OpenAI) multi-modal CLIP ViT-B/32 model. We use zero-shot ImageNet-1K accuracy as our pre-trained metric and then train a classification head on top of the image encoder of the CLIP model. We follow a similar experimental setup to section 5.1 in the paper, fine-tuning on 7 different downstream tasks. Due to a lack of space, we are omitting the rest of the details here. Please see the results below.
>
> | Method                        | IN-1K (Top-1 Acc) | Target Acc | Average |
> |-------------------------------|-------|------------|---------|
> | Pre-trained                   | 61.66 | 64.76      | 63.21   |
> | Standard FT                 | 1.98   | 88.80      | 45.39   |
> | Linear probe                 | 61.66 | 84.84      | 73.25   |
> | **FLOW (Ours)**          | 60.76 | 90.53      | **75.64**   |
>
> *FLOW performs the best by a decent margin*.
>
> **Distillation paper [1]:** Based on the feedback, we compared the distillation-based idea (abbreviated as LwF in [1]) against ours for the new ViT-B/16 experiment on Food101. We used the hyper-parameters for LwF suggested by [1]. We observe that our method FLOW outperforms LwF despite the simplicity; FLOW achieves better performance on the forgetting front, while LwF has higher accuracy on the fine-tuning task.
>
> | Method                 | IN-1K (top-1) | Food101 | Average |
> |------------------------|---------------|---------|---------|
> | Pre-trained           | 81.10         | ---     | ---     |
> | Standard FT          | 56.11         | 91.60   | 73.86   |
> | LwF [1]         | 76.39         | 91.23   | 83.81   |
> | **FLOW (Ours)**                | 77.94         | 90.57   | **84.26** |
>
> We believe our method has clear advantages on the efficiency front as well. Note that **the LwF method comes with more tunable parameters, additional memory, and evaluation overhead**. They have a distillation loss scaling factor $\lambda_o$ and “softening parameter” $T$ that rescales the outputs as $y’_i = (y’_i)^{1/T} / \sum_k^l (y’_k)^{1/T}$ for an $l$-dimensional output vector. In contrast, our method only has $\tau$, for which we use a prescribed value. Second, the logits obtained per sample for the pretrained model body and head must be stored to compute the distillation loss, which is infeasible in the language modeling setting. *Our method shows comparable performance with easier tuning and better storage efficiency*. Thank you for pointing out this line of work. Combining the distillation idea with our sample selection idea is also an interesting future work.
>
> We hope to have resolved your concerns, and we're happy to discuss further. *If you’re satisfied, we sincerely hope you will raise your score*!
>
> ----
>
> [A]: Wang et al., 2025. “LiNeS: Post-training Layer Scaling Prevents Forgetting and Enhances Model Merging”
>
> [B]: Wortman et al., 2022. “Robust fine-tuning of zero-shot models”

---

### Official Review · Reviewer_QV3i · 2025-03-13

**Overall Recommendation:** 3

**Summary:**

This paper examines how we can mitigate catastrophic forgetting during fine-tuning. They propose that prioritizing "easy" examples (i.e. those that the base model already has low loss on). Concretely, this allows mitigating forgetting without significant access to the pretraining dataset -- as would be required by a replay-based setup. The authors test their method on an Imagenet based vision task, as well as fine-tuning large language models for mathematical reasoning. Additionally, the authors demonstrate that their method has compositional benefits when combined with other robust fine-tuning approaches (such as WISE-FT). Finally, this paper examines a theoretical setup which formalizes the gains from prioritizing easier examples, by preventing overfitting along noise directions of the finetuning dataset.

**Claims And Evidence:**

The author's principal claim is that prioritizing easy examples via their loss weighting can enable a better tradeoff between fine-tuning in distribution performance and mitigating catastrophic forgetting from fine-tuning. They support their claim using an Imagenet finetuning task and a mathematical reasoning task. In general, the experiments that were conducted were well done and the authors compare to a variety of existing robust finetuning methods including full-ft, weight interpolation, and regularization. The results in vision setting wee especially remarkable with massive reductions in the forgetting while maintaining a lower, but comparable in-distribution performance. The results in LLM setting experienced smaller level of forgetting, but seemed to have similar relative trends. While the experimental evaluations are a promising start, the experiments are relatively limited and it would be nice to see a wider range of fine-tuning tasks represented. Particularly in the language modeling setting, for example, the claim would be better supported if also established on common fine-tuning tasks such as instruction following, safety training, etc. In particular, the authors make the particularly strong claim that "FLOW strikes a good balance between learning a new task and retaining knowledge from pre-training". However, I find that (particularly for the language modeling setting), this cannot be established simply from one mathematical reasoning task. Rather, the claim would be better supported by a suite of tasks that better leverage a wider section of the LLMs capabilities (such as tasks requiring factual/world knowledge, tasks requiring simply linguistic capabilities, etc).

**Essential References Not Discussed:**

As I mentioned above, the results are highly related to findings in language model factuality and it would be good for the authors to discuss these relationships [1,2].


[1] Ghosal, Gaurav, Tatsunori Hashimoto, and Aditi Raghunathan. "Understanding finetuning for factual knowledge extraction." arXiv preprint arXiv:2406.14785 (2024).

[2] Gekhman, Zorik, et al. "Does fine-tuning LLMs on new knowledge encourage hallucinations?." arXiv preprint arXiv:2405.05904 (2024).

**Experimental Designs Or Analyses:**

In general, I did not identify any major weaknesses of the experimental designs.

**Methods And Evaluation Criteria:**

Generally, the authors did a good job of selecting appropriate baseline methods which does make their result interesting and compelling. As I mentioned in "Claims and Evidence", the number of finetuning tasks/benchmarks sampled could be made a bit more diverse to comprehensively support their claim. Of course, the current math finetuning setting is an important component of this.

**Other Comments Or Suggestions:**

Not applicable

**Other Strengths And Weaknesses:**

Overall, I do think this is a strong paper that taps into a somewhat unexplored topic of how finetuning data point choice affects learning and forgetting. While prior works have studied this more specifically in factuality and knowledge setting, a more general characterization is not present to my knowledge. While limited, the experimental results are promising. Additionally, the theoretical results are thought provoking and well done in my opinion.

**Questions For Authors:**

1) Can authors also run their method on finetuning tasks such as instruction following, safety refusal, or context-based question answering for the LLM setup. Having some additional finetuning tasks for LLMs would greatly strengthen the paper.

2) Can authors contextualize/discuss the relationship between their results and prior works on factuality and the effect of training with unfamiliar/unpopular facts?

**Relation To Broader Scientific Literature:**

Robust fine-tuning and catastrophic forgetting are crucial issues in the foundation model era. As authors mention, there have been multiple classes of approaches to including (regularization/capacity constrained finetuning and replay based techniques). In this work, the authors make the very valid observation that replay-based techniques are infeasible given the closed nature of pretraining corpora and also the scale. This paper makes an exciting observation that rather than carefully craft novel fine-tuning algorithms and regularizers we can simply reweight the finetuning data to mitigate forgetting. This result dos appear novel in my view. However, it does reflect certain results found in the knowledge-based question answering setting about the role of different data points in finetuning [1,2]

**Theoretical Claims:**

I briefly examined the theoretical claims and they appear reasonable to me...haven't carefully checked correctness.

---

> ### Author Rebuttal · Authors · 2025-04-01
>
> We thank the reviewer for recognizing the novelties of the simple design of our sample-wise weighting scheme. We appreciate the constructive and encouraging feedback. Please find our responses to your major comments below.
>
> **Wider range of fine-tuning tasks, especially for language modeling:** While we agree that our language modeling section could benefit from additional fine-tuning domains (e.g., instruction tuning, question answering, safety alignment, etc.), we are unfortunately unable to present any further experiments due to compute and time constraints. More specifically, for example, we don’t have LLM-as-a-judge set up to evaluate instruction tuning.
>
> We appreciate you pointing out several of these fine-tuning domains, but we would like to clarify that *we did not intend to focus on this method for specific language tasks*. Rather, we have proposed this method oblivious to any particular architectural or data modality, with theoretical intuitions for linear models. However, these are great suggestions for our algorithm’s applications – with potential adaptations – to specific important language tasks. Thanks!
>
> We additionally include results on fine-tuning the image encoder of a pre-trained (OpenAI) multi-modal CLIP ViT-B/32 model. We use zero-shot ImageNet-1K accuracy as our pre-trained metric and then train a classification head on top of the image encoder of the CLIP model. We present their results in the table below. As you see, FLOW performs the best even here by a decent margin.
>
>
> | Method                        | IN-1K (Top-1 Acc) | Target Acc | Average |
> |-------------------------------|-------|------------|---------|
> | Pre-trained                   | 61.66 | 64.76      | 63.21   |
> | Standard FT                 | 1.98   | 88.80      | 45.39   |
> | Linear probe                 | 61.66 | 84.84      | 73.25   |
> | **FLOW (Ours)**          | 60.76 | 90.53      | **75.64**   |
>
> We follow a similar experimental setup to section 5.1 in the paper, fine-tuning on 7 different downstream tasks. Due to a lack of space, we are omitting the rest of the details here.
>
> **Relationship to prior works on factuality (Ghosal et al., 2024; Gekhman et al., 2024) and the effect of training with unfamiliar/unpopular facts:** This is an interesting perspective regarding our weighting scheme and on what basis the samples could be “easy” or “hard”. We came up with FLOW with a more general perspective and problem scope in mind, without a specific consideration for language modeling and factuality. However, we do see the connection. Thank you for bringing it to our attention.
>
> Within the context of factuality and LLMs, for a fine-tuning task on some factual knowledge-based dataset, which has samples that are not properly represented in the pre-training distribution, the respective samples could be considered hard, and vice versa. In a data-oblivious setting, as we do not have access to pre-training data, one way (probably the only principled way) would be to rank hardness based on pre-trained losses. Intuitively, such “hard” samples would have high losses, and our algorithm would give them smaller weights. Given the apparent connections, FLOW should intuitively behave in parallel with the findings of (Ghosal et al., 2024; Gekhman et al., 2024). We will include these discussions in the next version of our paper.

---

### Official Review · Reviewer_Ej68 · 2025-03-15

**Overall Recommendation:** 4

**Summary:**

The work relates to the issue of catastrophic forgetting in finetuned models wherein the pre-trained knowledge of the model is wiped out substantially after finetuning, and proposes a method that up-weighs the samples in the finetuning dataset that incur a low loss value (in contrast with existing approaches), in order to retain the pre-training knowledge. The up-weighing scale is theoretically determined and fixed at the beginning of pre-training and the method shows promising improvements over the baselines on known vision and language benchmarks. Theoretical analyses are provided to compare against vanilla finetuning under linear task regimes to evaluate convergence behavior.

**Claims And Evidence:**

The claims made in the submission are supported by clear and convincing evidence.

**Essential References Not Discussed:**

Could incorporate machine unlearning literature where finetuning is actively investigated.

**Experimental Designs Or Analyses:**

Yes, the experimental designs are sound. The experiments augmenting existing methods with FLOW are convincing of the method's merits. The work could benefit from an ablation for the choice of $\tau$ as the median value of the per-sample losses.

**Methods And Evaluation Criteria:**

The proposed methods and evaluation criteria are appropriate for the problem at hand.

**Other Comments Or Suggestions:**

Minor typo in line 247. LP and Linear Probing were both mentioned in Table 8.

**Other Strengths And Weaknesses:**

The work is original, thorough and simple. The manuscript is well written and structured. The work could benefit from a comparison between adaptive and fixed weighing scenarios and also from multi-task finetuning experiments.

**Questions For Authors:**

1. Could the work benefit further in terms of performance and efficiency if it incorporated a saliency mask in the pre-trained network inspired by [1].

2. Could the work be re-purposed for machine unlearning whereby there exists a "forget set" that the network is tasked for forgetting in addition to retaining it performance over the "retain set"? It could be a significant contribution to the unlearning literature if demonstrated.

3. How would the method handle outliers if the weights assigned to them were low based upon the loss values?

4. Is it possible to justify the choice of $\tau$ through a small scale ablation?

5. When applied to larger scale dataset/model scenarios, how may the method perform?

6. How would one approach a finetuning task using this method if the task is fairly/completely detached from the pre-training task? For instance, if finetuning a generative model over a classification task. Would the values for the sample losses be meaningful?




[1] Fan, Chongyu, et al. "Salun: Empowering machine unlearning via gradient-based weight saliency in both image classification and generation." arXiv preprint arXiv:2310.12508 (2023).

**Relation To Broader Scientific Literature:**

The work is pertinent to the finetuning and catastrophic forgetting literature as mentioned in the manuscript. Additionally the method may be relevant to machine unlearning literature [1, 2] wherein weighing samples from the retain/forget sets using the method in addition to masking based strategies [1] could enhance performance.


[1] Fan, Chongyu, et al. "Salun: Empowering machine unlearning via gradient-based weight saliency in both image classification and generation." arXiv preprint arXiv:2310.12508 (2023).

[2] Jia, Jinghan, et al. "Model sparsity can simplify machine unlearning." Advances in Neural Information Processing Systems 36 (2023): 51584-51605.

**Theoretical Claims:**

The proofs were checked for Appendix C, D, E.

---

> ### Author Rebuttal · Authors · 2025-04-01
>
> Thanks for the positive assessment of our work! We address your questions below.
>
> **Ablation for $\tau$:** In the table below, we show the pre-training and fine-tuning accuracies as a function of $\tau$ when the model is ResNet-50, the pre-training dataset is ImageNet-1K, and the fine-tuning dataset is Caltech 101. The last column represents the percentile of pre-training losses, which we use as the value of $\tau$.
>
> | **Pre-trained IN-1K (Top-1 Acc)** | **Fine-tuning Caltech101 (Acc)** | **Percentile (%)** |
> |--------------|------------|----------|
> | 68.51       | 91.15      | 10         |
> | 64.13       | 91.01      | 30         |
> | 54.72       | 91.80      | 50         |
> | 45.59       | 92.91      | 70         |
> | 20.51       | 94.02      | 90         |
>
> In our experiments, we used $\tau = 50^\text{th}$ percentile. While smaller $\tau$ works better here, in general, it seems very difficult to “guess” the optimal value of $\tau$ without any access to the pre-training data. But this is an interesting point which we will investigate in the future. Thanks for bringing this up!
>
> **Regarding Machine Unlearning (MU):** Thanks for pointing out papers on MU! [1, 2] fall within the more prominent category of weighted updates in the parameter space. As we discussed in our paper, there are several ideas in the parameter space for mitigating forgetting. *Our approach is orthogonal to such ideas as we perform weighting in the sample space*. One could extend the ideas of [1, 2] for forgetting and apply them with our sample-weighting approach (as we did with existing baselines in Section 6.2). Similarly, as you pointed out, one could extend our sample-weighting scheme for MU. These are great future directions! However, it is worth mentioning that while MU and forgetting sound similar, they are not the same in spirit; so it may not be straightforward to extend techniques of one into another. In MU, we deliberately (and ideally, provably) want to induce “forgetting” on some samples, whereas in our context, forgetting is an undesirable side effect that we want to avoid as much as possible. However, we will discuss the connections between MU and [1,2] in the next version of our paper.
>
> **Comparison between adaptive and fixed weighing scenarios and also from multi-task finetuning experiments:** These are great points! We opted for the fixed weighting scheme due to two reasons: (a) a principled adaptive scheme in the absence of pre-training data was not clear to us (specifically, in the way we derived Alg. 1 in Section 4), and (b) to keep the fine-tuning procedure light-weight. The multi-task finetuning setting may require a bit more thought, especially when the tasks appear sequentially – specifically, how the weights are chosen for each new task. This is left for future work.
>
> **Questions For Authors:**
>
> **1.** Potentially, yes! As we discussed above, a combination of our approach and the approach of [1] may lead to better performance.
>
> **2.** This is a great direction! Potentially, our sample weighting scheme could be extended to the machine unlearning problem.
>
> **3.** If there are *out-of-distribution* outliers with large pre-training losses, then our weighting scheme would assign them low weights and so the influence of such outliers would be curtailed. Hence, our algorithm handles such bad samples. On the other hand, if by outliers, you mean in-distribution samples with large pre-training losses, then it seems difficult to do well on such samples in general if we wish to also do reasonably well on the pre-training data, especially without any access to the pre-training data. The premise of our work is that such samples should be considered detrimental to the pre-training performance as they’d make the model drift away from the pre-training weights.
>
> **4.** Answered at the top.
>
> **5.** We believe our language experiments are on reasonably large-scale models and datasets. As for vision, please see our response to Reviewer 1DND, where we show additional results with (i) a ViT-B/16 model on Food101 which is much a harder dataset with 101k images, and (ii) CLIP ViT-B/32 (OpenAI) which is pre-trained with the contrastive loss but we fine-tune the image encoder with a cross-entropy loss.
>
> **6.** The premise of this work is that there is some kind of alignment between the pre-training and fine-tuning tasks; otherwise, it is very hard to find a model that does well on both tasks. Moreover, we assume that the pre-training losses are partially reflective of the “hardness” of the fine-tuning examples; otherwise, in the absence of any pre-training data, this seems like a very hard problem. In the case of a pre-trained generative model, the extension should be straightforward. Consider BERT, which is trained using masked language modeling. We would do linear probing of the [CLS] token’s output embedding, followed by applying Algorithm 2 (Appendix B).

---

### Official Review · Reviewer_txXi · 2025-03-16

**Overall Recommendation:** 4

**Summary:**

The paper contributes the following:
 - a weighting strategy (called FLOW) for finetuning examples in the data-oblivious setting where there is no access to pre-training examples, that upweights examples with small loss, and aimed at mitigating forgetting of the pretraining task.
 - a benchmark against other strategies on vision and language tasks, which demonstrates a higher mean accuracy between the pre-training task and fine-tuning task, or put differently, less forgetting.
 - a theoretical analysis in a linear setting trained by minimizing the population l2 loss (meaning access to infinite data) using gradient descent with constant learning rate.

**Claims And Evidence:**

The theoretical claims are rigorously prooved as far as I can tell. They apply to a particular toyish setup, so the empirical evaluation on actual data is welcome.

**Essential References Not Discussed:**

Not applicable.

**Experimental Designs Or Analyses:**

As state above, I would appreciate a discussion regarding the stopping criterion for the fine-tuning task, which is in effect a tradeoff between pretraining accuracy and finetuning accuracy.

**Methods And Evaluation Criteria:**

The chosen benchmark are particularly relevant. A question comes to mind however: how is the trade-off chosen between carrying on fine-tuning, vs stopping at some point. The number of fine-tuning iterations looks as a very important hyperparameter, whose choice should be discussed in more details.

**Other Comments Or Suggestions:**

Not applicable.

**Other Strengths And Weaknesses:**

The paper is clearly written and comprehensive, yet compact. I appreciate the fact that the proposed method and experiments are supported by some analytical results, even in a simplified setting.

Unless any other reviewer finds a major flaw in the methodology or proofs, I recommend acceptance.

**Questions For Authors:**

1. What does the distribution of example weights actually look like in your experiments? Is it concentrated on only a few examples?

Some questions that came to mind while reading your paper,
 2. Is the accuracy on minority examples (i.e. rare instances or "difficult" sub-patterns in the distribution, think i.e. fairness related issues) lower when finetuning using your training strategy, since corresponding examples would probably be downweighted.
 3. What happens when the pretraining and finetuning tasks are very different, and all finetuning examples get a high loss ?

**Relation To Broader Scientific Literature:**

The previous literature was appropriately cited.

**Theoretical Claims:**

I skimmed through the proof which looked correct to me, and the exposition is clear and easy to follow through.

Overall, this part however lacks a more thorough discussion of the setup. What is implied by this particular choice of distribution on x, and on y ? Is it somehow representative of actual tasks ? I understand that this specific choice of covariance matrix \hat{\Sigma} what chosen because it makes it possible to derive analytical training dynamics, but how does it translate to the relationship between the pre-training and fine-tunins tasks in actual settings ?

---

> ### Author Rebuttal · Authors · 2025-04-01
>
> Thanks for the positive assessment of our work! We address your questions below.
>
> **Number of fine-tuning epochs:** For the language experiments, this was chosen based on existing literature recommendations for fine-tuning epochs. Following the observation of [1], we trained for 2 epochs due to compute constraints. For vision, we selected the number of epochs to achieve a fine-tuning accuracy comparable to that of standard fine-tuning on specific models and tasks, aligning with the performance reported in [2].
>
> **Theoretical Claims:**
> Our theory setting is a typical linear setting, commonly used for analysis purposes. However, as mentioned in footnote 2, our insights carry over to wide neural networks following the dynamics of linear models under gradient descent (Lee et al., 2019). For a general choice of $\widetilde{\Sigma}$, we derived the weighted covariance matrix $\widetilde{\Sigma}'$ impacting the dynamics of FLOW in Appendix E (see eq. 32). Note that $\widetilde{\Sigma}'$ depends on $\widetilde{\Sigma}$ and $e = \theta_{\ast} - \widetilde{\theta}_{\ast}$; the latter is the difference between the optimal solutions of the pre-training and fine-tuning tasks.
>
> Unfortunately, as we explained in Remark E.1, it is hard in general to characterize the eigen-spectrum of $\widetilde{\Sigma}'$. But informally, *more is the alignment between $e$ ($= \theta_{\ast} - \widetilde{\theta}_{\ast}$) and the eigenvectors of $\widetilde{\Sigma}$, slower the convergence along a direction that is aligned to $e$, inhibiting overfitting to the fine-tuning data*. Formalizing this intuition is left for future work.
>
> **Questions For Authors:**
>
> **1.** Please see Fig. 3 (in Appendix I) for the distribution of sample weights of Gemma 2 2B, when we train with sequence-wise weights (results in the main paper) as well as token-wise weights (which we did as an ablation in the Appendix). Note that the distribution is Gaussian-like in the sequence-wise case, while it is concentrated in the token-wise case; as a result, the latter didn’t work very well. We’ll also produce similar plots for our vision experiments in the future.
>
> **2.** Indeed, the samples with high pre-training losses have lower accuracy when using FLOW compared to standard fine-tuning (FT).
>
> | Datasets       | Samples with top 10% highest losses | Standard FT | FLOW (Ours) |
> |---------------|-----------------|---------|-------|
> | CIFAR10       | 1000            | 86.60   | 30.70 |
> | CIFAR100      | 1000            | 56.40   | 21.30 |
> | Stanford cars | 805             | 71.30   | 13.175 |
>
> In fact, this behavior is in line with the premise of our approach, namely, sacrificing performance on the hard examples from the fine-tuning data to maintain performance on the pre-training data.
>
>
> **3.** If the fine-tuning examples have high losses with the pre-trained model ($\theta_{\ast}$), it probably indicates that there is no solution in the vicinity of $\theta_{\ast}$ for which the performance on the fine-tuning data is good. Without any access to the pre-training data, all we can hope to do is to remain "close" to $\theta_{\ast}$ to have decent performance on the pre-training data. In that case, all the methods that we are aware of should struggle to achieve good performance jointly on pre-training and fine-tuning data. Regarding the distribution of the weights, the value of the pre-training losses doesn’t really matter; what matters is their relative spread because the temperature $\tau$ is chosen as the median of the pre-training losses.
>
> ---
>
> [1]: Biderman et al., 2024. “LoRA Learns Less and Forgets Less.”
>
> [2]: Wightman et al., 2021. “Resnet strikes back: An improved training procedure in timm.”

---

### Decision · Program_Chairs · 2025-05-01

**Decision:**

Accept (spotlight poster)

**Comment:**

This paper tackles fine-tuning, and argues it is important to both perform well on the pre-train dataset and the new finetune dataset, but without access to the dataset used during pre-training. They propose to upweight 'easy samples' in order to achieve better performance.

Reviewers all agree to accept the paper, and the authors rebutted many of the points reviewers raised in a convincing manner. Therefore this publication should be accepted.

Reviewers agree that this paper is well-written, the experiments are good, and the setup is important. There were concerns about not enough image experiments, but additional experiments were added during the rebuttal.

Two reviewers (txXi and Ej68) brought up the point about outliers / fairness issues of forgetting 'minority populations' when using this technique. I expect the authors to comment on this in a limitations and in the broader impacts. I trust the authors will do this given they added results already in the rebuttal.

Reviewer 1DND brings up Learning without Forgetting. In general, this fine-tuning setup (not forgetting the pre-train data) is very similar to a 2-task continual learning setting. The authors have added LwF results (although it is not clear what hyperparameter selection procedure was used), and in general refer to continual learning literature in their related works. However, their other baselines are very simple and do not consider continual learning. Typically, continual learning methods (like LwF) perform very well in a 2-task setting because it is easy to tune the hyperparameter to trade-off forgetting vs learning. That said, a benefit of this work may be that there is little-to-no tuning of hyperparameters required.